# HOW DOES FINE-TUNED FOUNDATION MODELS HELP FOR LONG-TAILED DATA

## ABSTRACT

Deep long-tail learning is a challenging visual recognition problem that trains models on long-tailed distributed datasets. In the last decade, a large number of methods have been proposed to solve the problems caused by imbalanced data. Many methods have been proven useful in learning a deep model from scratch, such as ResNet or ResNeXt, but they have not been validated as effective in fine-tuning the pre-trained foundation models, such as CLIP or ViT. If users inappropriately apply these long-tail learning methods, it may result in worse accuracy than expected. However, there is no scientific guideline for these methods in the existing literature. In this paper, we first collect the widely used methods of existing long-tail learning and then conduct extensive and systematic experiments to provide a guideline for the accurate use of these methods in fine-tuning foundation models. Furthermore, we observe that the current comparison protocol ignores the influence of training cost and hyperparameter selection, which may potentially lead to unfair comparisons and biased results. Motivated by our empirical studies, we propose a unified fine-tuning framework for long-tailed recognition. Experimental results demonstrate that the proposed framework outperforms existing methods on multiple long-tailed datasets, including ImageNet-LT, Places-LT, CIFAR100-LT, and iNaturalist 2018.

## 1 INTRODUCTION

Deep neural networks have achieved great success in a variety of computer vision tasks, such as image recognition (Voulodimos et al., 2018; Krizhevsky et al., 2012), object detection (Zhao et al., 2019; Zou et al., 2023), etc. These achievements are attributed to the availability of large-scale datasets (Deng et al., 2009; Zhou et al., 2017; Krizhevsky, 2009) and the elaborately designed models (He et al., 2016; Dosovitskiy et al., 2021). However, in the real world, the natural data typically exhibits a long-tailed distribution (Liu et al., 2019; Cao et al., 2019; Kang et al., 2020; Yuan et al., 2021a; Yan et al., 2023; Xu et al., 2023a), where a small number of head classes have the majority of samples, and a large number of tail classes have only a few samples. Such extreme class imbalance poses severe challenges to the training of deep neural networks. The reason lies in that the models are prone to making predictions biased towards the head classes, leading to poor performance on tail classes, thereby decreasing the overall prediction performance (Tan et al., 2020; Zhang et al., 2023).

To solve the long-tail problem, many methods have been proposed in recent years. For example, re-weighting methods (Wu et al., 2020; Khan et al., 2019; Cui et al., 2019) aim to adjust the training loss for each class by multiplying it with a different weight; re-sampling methods (Chawla et al., 2002; Liu et al., 2008; Shi et al., 2023) aim to adjust the number of samples for each class in each sample batch to rebalance the classes; ensemble learning methods (Zhou et al., 2020; Wang et al., 2021b) aim to combine multiple exports to reduce the bias of the model towards the head classes. These existing methods have made significant progress in improving classification accuracy, but the experimental results of these methods are obtained from models trained from scratch, with limited research on fine-tuning pre-trained foundation models.

Recently, some works study long-tail learning with foundation models instead of training from scratch, such as BALLAD (Ma et al., 2021), VL-LTR (Tian et al., 2022), LPT (Dong et al., 2023), LIFT (Shi et al., 2024), and RAC (Long et al., 2022). However, these studies are less comprehensive and lack a systematic investigation. BALLAD and VL-LTR focus on two-stage learning methods,

while LPT and LIFT utilize rebalanced loss functions to mitigate the long-tail problem. On the other hand, BALLAD, VL-LTR, and RAC only apply the full fine-tuning setting, while LPT and LIFT focus solely on the parameter-efficient fine-tuning approaches. To the best of our knowledge, there has not been a systematic study on how to fine-tune foundation models under a long-tailed distribution.

In this paper, we delve into the commonly used methods in long-tail learning and apply them to fine-tune pre-trained CLIP (Radford et al., 2021) and ViT (Dosovitskiy et al., 2021), which are widely used in various visual tasks (Dehghani et al., 2023; Zhou et al., 2022; Yuan et al., 2021b; Wang et al., 2021a; Gao et al., 2024). We conduct extensive and systematic experiments to evaluate whether these methods are equally effective on foundation models as learning from scratch. We also analyze their training costs and hyperparameter selections. Finally, motivated by the results of our empirical studies, we integrate the optimal methods and propose a unified training framework. The proposed framework achieves better results than existing approaches on multiple long-tailed datasets, including ImageNet-LT (Liu et al., 2019), Places-LT (Sharma et al., 2021), CIFAR100-LT (Cao et al., 2019), and iNaturalist 2018 (Van Horn et al., 2018).

The main contributions of our work are as follows:

- We thoroughly explore the effectiveness of commonly used methods in long-tail learning when applied to foundation models to provide guidance for future research.
- We propose a unified fine-tuning framework by assembling optimal methods, which outperforms existing methods on multiple long-tailed datasets.
- We investigate training costs and hyperparameter selection in experiments to offer comprehensive recommendations for the use of these methods in practical settings.

## 2 RELATED WORK

**Long-Tail Learning** There are several methods being proposed to address the long-tail problem (Liu et al., 2019; Cao et al., 2019; Cui et al., 2019; Kang et al., 2020; Zhou et al., 2020; Zhong et al., 2021; Yang et al., 2022; Zhang et al., 2023), which can be divided into three categories (Zhang et al., 2023): 1) Class re-balancing aims to enhance the model's ability to recognize minority classes by rebalancing the sample proportions across different classes, including re-sampling (Chawla et al., 2002; Liu et al., 2008; Shi et al., 2023), class-sensitive re-weighting (Wu et al., 2020; Khan et al., 2019; Cui et al., 2019), and logit adjustment (Menon et al., 2021; Zhang et al., 2021a; Hong et al., 2021). 2) Information augmentation aims to improve model performance on long-tailed data by incorporating additional information during model training, including transfer learning (Cui et al., 2018; Xiang et al., 2020) and data augmentation (Shorten & Khoshgoftaar, 2019; Zhong et al., 2021). 3) Module improvement methods seek to address long-tail problems by improving network modules or representations, including classifier design (Wu et al., 2021; Liu et al., 2021a), contrastive learning (Kang et al., 2021; Zhu et al., 2022), and ensemble learning (Zhou et al., 2020; Wang et al., 2021b). However, these works only study how to train models from scratch and ignore the development of pre-trained foundation models. In this paper, we aim to further investigate the specific effects of the representative methods by applying them to the advanced foundation models.

**Fine-Tuning Foundation Models** The pre-trained foundation models have attracted widespread attention in recent years (Vaswani et al., 2017; Dosovitskiy et al., 2021; Radford et al., 2021; Touvron et al., 2021; Liu et al., 2021b). These models are pre-trained on web-scale data to construct sophisticated features and transferred to various downstream tasks, such as image classification (Yuan et al., 2021a), object detection (Yan et al., 2023), and semantic segmentation (Xu et al., 2023a). Moreover, the adaptation to downstream tasks can be further improved by applying extra data to fine-tune the foundation model (Dosovitskiy et al., 2021; Zhou et al., 2022). There are two fine-tuning approaches: full fine-tuning (Kumar et al., 2022) and parameter-efficient fine-tuning (Zaken et al., 2022; Jia et al., 2022; Chen et al., 2022), where the latter is regarded as a typical efficient mode by introducing only a few learnable parameters. However, these methods mainly utilize the balanced data for fine-tuning, which may yield unsatisfactory results when directly applied to the long-tailed datasets (Shi et al., 2024). Although some works have been proposed to mitigate this issue (Ma et al., 2021; Tian et al., 2022; Dong et al., 2023; Zhang et al., 2021b), no research has systematically studied the impact of long-tail learning algorithms on foundation models. For the

first time, we explore the reasonable application of long-tail learning methods on foundation models to provide a guideline for future applications.

## 3 METHODS GALLERY

We commence by introducing the Problem Definition, then categorize classical long-tail learning methodologies into 7 distinct groups: 1) Re-sampling, 2) Data Augmentation, 3) Class-sensitive Loss, 4) Balanced Classifier, 5) Knowledge Distillation, 6) Ensemble Learning, and 7) Other tricks. For each group, we first revisit relevant methods and then compare experimental performance. To ensure the reliability of our investigation, we experiment under different scenarios, including different foundation models (CLIP and ViT) and different fine-tuning paradigms (FFT and PEFT). Comprehensive details regarding the datasets and implementation settings are provided in Appendix Section A. Due to the page limit, the knowledge distillation method is introduced in Appendix Section B, and the ensemble learning method is presented in Appendix Section C.

### 3.1 PROBLEM DEFINITION

Long-tailed recognition aims to learn deep classification models from training datasets characterized by a long-tailed class distribution, where a small number of classes contain a large number of samples, while the majority of classes have only a few samples. Formally, we denote the long-tailed datasets with $N$ samples as $D = \{x_i, y_i\}_{i=1}^N$. Besides, we denote $n_i$ as the sample frequency of class $i$ ($1 \leq i \leq K$), then we have $N = \sum_{i=1}^K n_i$. In long-tail learning, the class frequencies are arranged in a descending order (Kang et al., 2020), i.e., if $1 \leq i < j \leq K$, then $n_i \geq n_j$. The imbalance ratio is defined as $r = \frac{n_1}{n_K}$, representing the ratio between the class with the largest number of images and the class with the smallest number of images, which can be used to describe the severity of the long-tailed distribution. In practice, $r$ formulates a large number, which indicates that $n_1 \gg n_K$ in a long-tailed dataset. The goal of long-tail learning is to learn a model $M$ from the imbalanced data $D$ so that $M$ can attain optimal predictions on test data.

### 3.2 RE-SAMPLING

Due to the intrinsic data imbalance in the long-tailed data, conventional sampling methods result in more head-class samples than tail-class samples in each training batch (Kang et al., 2020; 2021; Zhu et al., 2022). Re-sampling tackles this issue by adjusting the sample distribution of each class within the training data.

**Re-sampling Methods**  We investigate several classic and widely used re-sampling methods.

- Random Over-Sampling (ROS) (Buda et al., 2018) balances the data distribution by duplicating samples from the tail classes to increase their proportion in training data to achieve a more balanced sample distribution between head classes and tail classes.

- Random Under-Sampling (RUS) (More, 2016) aims to balance the data distribution by reducing the number of samples from the head classes to make their sample frequencies closer to those of the tail classes.

- Equalized re-sampling (EQ) (Kang et al., 2020; Shi et al., 2023) dynamically applies over-sampling or under-sampling to different classes by ensuring the total size of the dataset is unchanged. In this case, it obtains a balanced dataset without adding more training overhead.

- Square-root sampling (Kang et al., 2020) addresses limitations of balanced re-sampling—excessive discarding of head-class samples and redundant duplication of tail-class samples. This approach samples class $j$ with probability $p_j = \frac{n_j^q}{\sum_{i=1}^K n_i^q}$ ($n_i$ = class sample count). Setting $q = \frac{1}{2}$, it reduces head-class sampling frequency while preventing over-balancing between head and tail classes.

**Experimental Result**  Table 1 shows the results of using different re-sampling methods on CIFAR100-LT and Places-LT datasets. For more detailed results, please refer to Appendix section D.1

Table 1: Accuracy of re-sampling methods. "Baseline" represents no resampling. **Bold** and underlined numbers represent the optimal and sub-optimal results, respectively; the same notations are applied to all tables below.

| Datasets | CIFAR100-LT | | | | Places-LT | | | |
|---|---|---|---|---|---|---|---|---|
| Backbone | CLIP | | ViT | | CLIP | | ViT | |
| | FFT | PEFT | FFT | PEFT | FFT | PEFT | FFT | PEFT |
| Baseline | 54.6 | 71.9 | 70.3 | 80.7 | 24.7 | 39.8 | 26.0 | 32.1 |
| ROS | 44.8 | 68.3 | 48.3 | 71.0 | 12.6 | 38.3 | 11.4 | 32.2 |
| RUS | 45.5 | **77.4** | 69.3 | **87.0** | **42.3** | **50.8** | **41.2** | **45.3** |
| EQ | 50.4 | 72.8 | 62.0 | 77.3 | 21.7 | 43.7 | 22.0 | 33.7 |
| Square-root | **56.8** | 76.4 | **76.0** | 84.4 | 37.1 | 47.5 | 32.6 | 39.7 |

Based on our experimental findings, these sampling methods consistently perform better under the PEFT setting than under the FFT setting. RUS and Square-root sampling are proven to be more effective strategies, which can significantly enhance performance by more than 5%. In contrast, ROS exhibits significant performance deterioration, which is due to the severe overfitting issue. The performance of EQ is between RUS and ROS.

Given that the model is already pre-trained, these results appear to be justifiable: a minimal amount of data is sufficient to fine-tune the model and improve its performance on long-tailed datasets. We conduct an additional experiment to verify this point. Specifically, we compare the balanced dataset obtained through the RUS with 2, 5, and 10 times larger variants. Table 2 reports the results on the CLIP-ViT-B/16 PEFT setting, showcasing that RUS performs better, particularly on tail classes. As the data amount grows larger, though the head-class performance slowly increases, the tail-class performance exhibits significant declines.

We evaluated the models with two RUS and RUSx10 to enable a deeper mechanistic analysis. We extract the features of **tail-class** test data from CIFAR100-IR100 using these two models and visualize the results using t-SNE, as shown in Figure 1a 1b. The ellipse is constructed using the eigenvectors and eigenvalues of the covariance matrix, derived from the data's mean and covariance, which define its orientation, major and minor axes, and center.

Figure 1: t-SNE visualization and classifier weight norms for RUS and RUSx10.

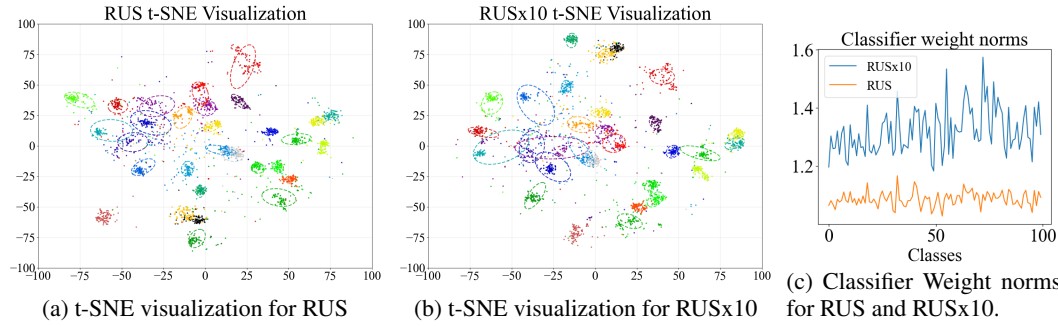

(a) t-SNE visualization for RUS     (b) t-SNE visualization for RUSx10     (c) Classifier Weight norms for RUS and RUSx10.

It can be observed that the ellipses in the t-SNE plot for RUS exhibit less overlap. This indicates more distinct decision boundaries for the tail classes, leading to better tail classification performance. Additionally, the weight norms of the model's classifier are presented in Figure 1c. RUS clearly demonstrates a more balanced distribution across all classes.

Furthermore, in terms of the training cost, the samples produced by RUS and Square-root sampling are significantly fewer, nearly 100 times less than those generated by ROS (the number varying with the dataset). Therefore, the training time cost is substantially lower than that of ROS and EQ under the same setting. Considering the above factors, using RUS or Square-root sampling is more practicable for fine-tuning foundation models with long-tailed datasets.

## 3.3 DATA AUGMENTATION

Data augmentation (Shorten & Khoshgoftaar, 2019) aims to increase data diversity by applying predefined transformations, thereby improving model generalization, especially in scenarios where the available data is limited.

Table 2: RUSxN indicates that the training dataset size is N times that of the RUS-sampled dataset, with each class containing N times the data as in RUS; "-" in the table means the corresponding experiment is not implemented due to the huge amount of data.

| Datasets | CIFAR100-LT | | | | Places-LT | | | |
|---|---|---|---|---|---|---|---|---|
| | Mean | Many | Med. | Few | Mean | Many | Med. | Few |
| RUS | **77.7** | 82.0 | 80.0 | **69.9** | **50.8** | 49.6 | 52.2 | **49.6** |
| RUSx2 | 77.5 | 85.3 | **80.6** | 64.6 | 50.6 | 50.5 | **52.6** | 46.1 |
| RUSx5 | 75.6 | 87.2 | 79.7 | 57.4 | 49.1 | 51.5 | 51.4 | 39.2 |
| RUSx10 | 73.4 | **88.1** | 77.8 | 50.7 | 47.7 | **52.7** | 48.9 | 33.9 |

Table 3: Accuracy of applying augmentation methods.

| Datasets | CIFAR100-LT | | | | Places-LT | | | |
|---|---|---|---|---|---|---|---|---|
| Backbone | CLIP | | ViT | | CLIP | | ViT | |
| | FFT | PEFT | FFT | PEFT | FFT | PEFT | FFT | PEFT |
| No augmentation | 48.7 | 71.9 | 71.1 | **81.6** | 23.7 | 39.8 | 25.7 | 31.7 |
| ColorJitter | 54.6 | 71.9 | 70.3 | 80.7 | 24.7 | 39.8 | 26.0 | 32.1 |
| RandAugment | 56.7 | **72.1** | 70.0 | 81.5 | **25.4** | 40.4 | 26.5 | 32.6 |
| AutoAugment | **57.8** | 70.7 | **71.6** | 81.3 | 24.9 | **40.7** | **26.9** | **32.7** |

**Augmentation Methods**   In our paper, in addition to conventional image processing, we apply several common data augmentation techniques.

- ColorJitter is one of the most commonly used methods for color-based data augmentation in images. It applies random transformations within a specified range to the image's brightness, contrast, saturation, and hue.

- AutoAugment (Cubuk et al., 2019) creates a search space of strategies, each containing multiple sub-strategies. For each mini-batch image, one sub-strategy is randomly selected. Each includes two processing functions—like rotation, inversion, or shearing—with their probability and magnitude parameters.

- RandAugment (Cubuk et al., 2020) is a simplified version of AutoAugment. The core of RandAugment is to randomly select a set of predefined augmentation operations with equal probability and assign an intensity hyperparameter to each operation to transform the input images.

**Experimental Results**   Table 3 shows the results of different augmentation methods on different datasets and settings. For more detailed results, please refer to Appendix section D.2.

Based on the experimental results, it can be concluded that solely applying data augmentation to long-tailed datasets can just slightly improve the performance of foundation models by less than 1%. Furthermore, when combined with other long-tail learning methods, data augmentation can not always gain benefits, which will be discussed in Section **The Ultimate Framework.**

Data augmentation introduces computational overhead during data preparation, consequently extending the total training duration. For example, our experiments demonstrate a 15% increase in end-to-end training time with RandAugment. In addition, we also research other impact of data augmentation on model training, as shown in Figure 2. We illustrate the convergence curves of training loss and accuracy for the ImageNet-LT dataset without augmentation and with AutoAugmentation. Based on the observations from the figures, it can be concluded that data augmentation slows down the convergence speed of the model. The reason why such kind of data augmentation without using external data faces difficulty in improving performance may be that foundation models have already seen various styles of images. Some recent studies have shown that introducing external data or knowledge for augmentation is effective (Long et al., 2022; Wang et al., 2024a), which may be an interesting direction in future research.

### 3.4 CLASS-SENSITIVE LOSS

Traditional deep learning methods typically employ the softmax cross-entropy loss function for training. However, this loss function often overlooks the issue of class imbalance among training data. We revisit some classic class-sensitive losses, which aim to rebalance the training loss for different classes to deal with the imbalance problem.

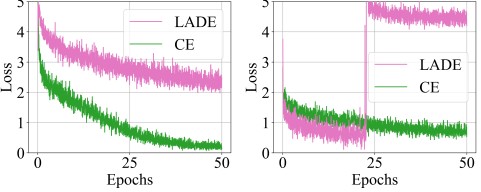

Figure 2: Convergence curves of training accuracy (left) and loss (right) on ImageNet-LT under CLIP-ViT-B/16 PEFT.

Figure 3: Training loss of LADE and CE on Places-LT under CLIP/B-16 FFT (left) and ViT/B-16 PEFT (right) setting

Table 4: Summary of losses. In the table, $z$ is the predicted logits, $p$ is the probability obtained by applying softmax to $z$, where $z_y, p_y$ correspond to class y. $\pi_y = \frac{n_y}{N}$ represents the label frequency of the class y, where $n_y$ represents the number of samples in class y, $N$ is the total sample numbers.

| Loss | Formulation | Hyperparam. | Loss | Formulation | Hyperparam. |
|------|-------------|-------------|------|-------------|-------------|
| CE | $-\log(p_y)$ | - | G-RW | $-\frac{(1/\pi_y)^\rho}{\sum_j (1/\pi_j)^\rho}\log(p_y)$ | $\rho$ |
| Focal | $-(1-p_y)^\gamma \log(p_y)$ | $\gamma$ | BS | $-\log(\frac{\pi_y exp(z_y)}{\sum_j \pi_j exp(z_j)})$ | - |
| LDAM | $-\log(\frac{exp(z_y-\Delta_y)}{\sum_j exp(z_j-\Delta_j)})$ | $s$ | LA | $-\log(\frac{exp(z_y+\mu\cdot\pi_y)}{\sum_j exp(z_j+\mu\cdot\pi_j)})$ | $\mu$ |
| CB | $-\frac{1-\beta}{1-\beta^{n_y}}\log(p_y)$ | $\beta$ | LADE | $L_{BS} + \alpha L_{LADER}$ | $\alpha, \lambda$ |

**Loss Functions**   We study common class-sensitive losses, which are listed in Table 4.

- **Focal Loss** (Lin et al., 2017): Modulates CE loss with $\gamma$ to down-weight easy examples.
- **LDAM** (Cao et al., 2019): Assigns class-dependent margins ($\Delta$) inversely proportional to class frequency.
- **CB Loss** (Cui et al., 2019): Reweights losses by the effective number of samples per class.
- **G-RW** (Zhang et al., 2021a): Generalizes re-weighting with scale parameter $\rho$.
- **Balanced Softmax** (Ren et al., 2020): Adjusts softmax weights by class sample sizes.
- **Logit-Adjusted** (Menon et al., 2021): Applies label-dependent offsets to logits based on class frequency.
- **LADE** (Hong et al., 2021): Calibrates outputs using test label distribution. Its regularizer $L_{LADER}$ combines class priors $\pi_j$ and normalization terms. $L_{LADER} = \sum_{j\in K} \pi_j L_{LADER_j}$, given $L_{LADER_j} = -\frac{1}{N_j}\sum_{i=1}^{N} \mathbf{1}_{y_i=j}\cdot\pi_j + Z + \sum_j \pi_j \lambda Z^2$, where $Z = \log(\frac{1}{N}\sum_{i=1}^{N}\frac{z_j}{K\pi_y})$.

**Experimental Result**   We present the experimental result in Table 5. For more parameter settings and results, please refer to Appendix section D.3.

Table 5: Accuracy of applying class-sensitive losses.

| Datasets | CIFAR100-LT | | | | Places-LT | | | |
|----------|------|------|------|------|------|------|------|------|
| Backbone | CLIP | | ViT | | CLIP | | ViT | |
|  | FFT | PEFT | FFT | PEFT | FFT | PEFT | FFT | PEFT |
| CE | 54.6 | 71.9 | 70.3 | 80.7 | 24.7 | 39.8 | 26.0 | 31.9 |
| Focal | 52.7 | 71.2 | 69.4 | 81.4 | 24.3 | 39.0 | 25.9 | 30.9 |
| LDAM | 53.6 | 73.6 | 64.4 | 82.8 | 24.7 | 41.1 | 25.0 | 30.9 |
| CB | 54.7 | 72.5 | 69.4 | 80.3 | 25.1 | 40.2 | 26.0 | 32.0 |
| G-RW | 50.9 | 71.8 | 66.9 | 81.8 | 22.0 | 44.5 | 23.4 | 34.2 |
| BS | 58.0 | **80.1** | **75.8** | 85.1 | **31.3** | 48.4 | 30.3 | 38.3 |
| LA | **62.7** | 79.8 | 73.1 | **86.3** | **32.0** | 48.0 | **31.9** | **39.7** |
| LADE | 18.2 | 79.9 | 72.8 | 86.0 | 16.8 | **49.2** | 27.3 | 0.3 |

In most cases, we find that Focal loss, Class-Balanced loss and Generalized Re-Weight loss achieve only moderate gains when applied to foundation models in both FFT and PEFT settings, and even impair the performance in some cases. LDAM loss shows a slight improvement only in the PEFT setting, with no improvement observed in the FFT setting. LADE loss is complex and highly sensitive to hyperparameter selection due to its two hyperparameters. We use the same parameters for

LADE across all experimental settings; however, in some cases, it provides a significant improvement, while in others, it leads to a notable performance drop and even causes training collapse. Figure 3 shows the training loss of the LADE under certain training settings, which fails to converge to lower values and even crashes during training, indicating the potential risk caused by improper hyperparameters. We believe the LADE loss function introduces numerous additional assumptions based on logit adjustment, making it overly complex. Therefore, it may only be suitable for specific models, such as CNN models, rather than ViT models.

In contrast, Balanced Softmax and Logit-Adjusted loss consistently proved to be effective methods for both FFT and PEFT in foundation models and can significantly improve model performance. Specifically, they sacrifice a little performance of the head class in exchange for significant improvements in the performance of the middle and tail classes. Based on the experimental results, we recommend using Balanced Softmax loss and Logit-Adjusted loss when fine-tuning foundation models with long-tailed datasets. If time spent on hyperparameter tuning is non-trivial, then the nonparametric BS loss is a more reliable choice.

## 3.5 BALANCED CLASSIFIER

In general visual tasks, a common practice in deep learning is to employ linear classifiers $p = \phi(w \cdot x + b)$ for classification, where $\phi$ is the softmax function, the bias term $b$ can be discarded. However, the long-tailed distribution data lead to larger classifier weight norms for head classes than tail classes (Yin et al., 2019). We investigate diverse classifier types to tackle this challenge.

**Classifier Methods**  We introduce two representative classifiers, *i.e.*, Cosine classifier and $\tau$-normalized classifier.

- Cosine classifier (Wu et al., 2021) uses a scale-invariant metric $p = \phi((\frac{w \cdot x}{||w|| \cdot ||x||})/t + b)$, in which both the classifier weights and the sample features are normalized. $t$ is the temperature parameter. This strategy can be motivated by removing the negative impact of imbalanced weight norms (Kang et al., 2020; Wei et al., 2021).
- $\tau$-normalized classifier (Kang et al., 2020) adjust the classifier weight norms to solve the imbalance by $\tau$-normalized procedure, typically used to enhance the performance and stability of models in high-dimensional data. Formally, $\tilde{w} = \frac{w}{||w||_2^\tau}$, where $\tau$ is temperature factor for normalization.

**Experimental Result**  In our experiments, we follow the setting of Shi et al. (2024) and Kang et al. (2020) and set the $t$ to $\frac{1}{30}$ in Cosine Classifier and $\tau$ to $0.5, 1, 2$ in $\tau$-normalized classifier. Table 6 shows the accuracy of different classifier methods on CIFAR100-LT and Places-LT datasets. For more detailed results, please refer to Appendix section D.4.

In our experiments, we observed comparable training costs across different classifiers. According to the experiment results, we can observe that in most cases, the Cosine classifier is a better choice because it has empirical robustness to imbalances and stronger generalization ability. Note that these classifiers are exclusive to each other and can't be used simultaneously. We recommend using the Cosine Classifier to train foundation models.

Table 6: Accuracy of applying different classifiers.

| Datasets | CIFAR100-LT | | | | Places-LT | | | |
|---|---|---|---|---|---|---|---|---|
| Backbone | CLIP | | ViT | | CLIP | | ViT | |
| | FFT | PEFT | FFT | PEFT | FFT | PEFT | FFT | PEFT |
| Linear | 54.6 | 71.9 | **70.3** | 80.7 | **24.9** | 39.8 | 26.0 | 31.9 |
| Cosine | **56.4** | **72.2** | 69.6 | **83.9** | 24.9 | **40.6** | **27.1** | **38.1** |
| $\tau$-norm ($\tau = 0.5$) | 55.6 | 71.7 | 69.3 | 80.8 | 24.7 | 40.3 | 25.8 | 32.1 |
| $\tau$-norm ($\tau = 1$) | 55.6 | 71.9 | 68.9 | 80.9 | 24.6 | 40.0 | 25.4 | 32.3 |
| $\tau$-norm ($\tau = 2$) | 54.8 | 71.8 | 68.8 | 81.2 | 23.5 | 37.6 | 24.8 | 32.1 |

## 3.6 OTHER TRICKS

In addition to the aforementioned methods, we also explore two more tricks: mixup (Zhang et al., 2018) and label smoothing (Szegedy et al., 2016), which are widely used in various types of deep models and long-tail learning algorithms (Zhong et al., 2021).

Table 7: Accuracy of applying mixup.

| Datasets | CIFAR100-LT | | | | Places-LT | | | |
|---|---|---|---|---|---|---|---|---|
| Backbone | CLIP | | ViT | | CLIP | | ViT | |
| | FFT | PEFT | FFT | PEFT | FFT | PEFT | FFT | PEFT |
| Baseline | 51.5 | **80.1** | 75.8 | 85.1 | 31.3 | 48.8 | 30.3 | 38.3 |
| Mixup | **68.7** | 79.7 | **81.6** | **86.7** | **35.8** | **49.8** | **33.3** | **45.0** |

Table 8: Accuracy of applying label smoothing.

| Datasets | CIFAR100-LT | | | | Places-LT | | | |
|---|---|---|---|---|---|---|---|---|
| Backbone | CLIP | | ViT | | CLIP | | ViT | |
| | FFT | PEFT | FFT | PEFT | FFT | PEFT | FFT | PEFT |
| CE | 54.6 | **71.9** | 70.3 | 80.7 | 24.7 | **39.8** | 26.0 | 31.9 |
| CE (w/ LS) | **56.2** | 71.7 | **71.3** | **82.7** | **25.0** | 39.7 | **26.9** | **34.1** |
| BS | 58.0 | 80.1 | 75.8 | 85.1 | **31.3** | 48.8 | 30.3 | 38.3 |
| BS (w/ LS) | **59.8** | **80.6** | **78.2** | **88.1** | 28.6 | **49.4** | **32.4** | **41.8** |

For the mixup trick, we follow the setting of Zhang et al. (2018). Specifically, we randomly select two data points $(x_i, y_i)$, $(x_j, y_j)$ from the original dataset and combine them through linear weighting. Formally,

$$\widehat{x} = \theta x_i + (1 - \theta)x_j \tag{1}$$
$$\widehat{y} = \theta y_i + (1 - \theta)y_j \tag{2}$$

where $\theta$ is randomly sampled from a Beta distribution $Beta(\zeta, \zeta)$. The mixup hyper-parameter $\zeta$ controls the strength of interpolation between feature-target pairs.

Label smoothing (Szegedy et al., 2016) transforms the training label from hard (one-hot) label to soft label, where the true label is considered to have a probability of $1 - \epsilon$, and the remaining $\epsilon$ is shared across all classes. After using label smoothing, the modified probability distribution is formulated as follows:

$$P_i = \begin{cases} 1, & \text{if } y = i \\ 0, & \text{if } y \neq i \end{cases} \Rightarrow P_i = \begin{cases} 1 - \epsilon, & \text{if } y = i \\ \frac{\epsilon}{K-1}, & \text{if } y \neq i \end{cases} \tag{3}$$

where $i$ is the $i$-th class, $K$ is the total number of classes and the hyperparameter $\epsilon$ determine the smooth level.

**Experimental Result** Table 7 and Table 8 show the test accuracy of using these two tricks. For more detailed results, please refer to Appendix section D.7.

For mixup, we set hyper-parameter $\zeta$ to 1. It can be observed that input mixup effectively provides better results compared to the baseline in both FFT and PEFT settings. Mixup can be seen as a form of data augmentation that combines multiple samples linearly, rather than applying transformations to a single sample. This linear behavior helps reduce the oscillations when the model predicts the out-of-distribution samples (Zhang et al., 2018). However, when combined with other long-tail learning methods, mixup may also not always gain benefits like those mentioned above in subsection Data Augmentation.

For label smoothing, we set the $\epsilon$ to 0.1 by the setting of Szegedy et al. (2016) and apply it to CE loss and BS loss. We find that label smoothing can effectively improve the final performance of CE loss and BS loss. More specifically, label smoothing enhances the performance of tail classes, as shown in tables 40, 41, 42 in the Appendix. Our results suggest the noise introduced by label smoothing effectively reduces the model's tendency to overly favor head-class samples, allowing for greater focus on tail-class samples.

## 4 THE ULTIMATE FRAMEWORK

**Framework construction** In the previous section, we review several classical methods. In this section, we analyze these methods from a more unified perspective. Specifically, we compare the different combinations of these methods to identify the best framework. It is worth noting that since re-sampling methods and class-sensitive losses both aim to re-balance the data distribution, their

Table 9: Results of the ablation experiments. "Avg." represents the average of all experimental results listed front in the line. $\Delta$ represents the performance change against the previous line. The abbreviations are defined as follows: "Cos" = Cosine Classifier, "Sqrt" = Square-Root Sampling, "BS" = Balanced-Softmax, "LS" = Label Smoothing, "Aug" = Auto Augmentation.

| Datasets | | | | | | ImageNet-LT | | | | iNaturalist 2018 | | | | | |
| --- | --- | --- | --- | --- | --- | --- | --- | --- | --- | --- | --- | --- | --- | --- | --- |
| Backbone | | | | | | CLIP | | ViT | | CLIP | | ViT | | **Avg.** | $\Delta$ |
| Cos | Sqrt | BS | LS | Aug | Mixup | FFT | PEFT | FFT | PEFT | FFT | PEFT | FFT | PEFT | | |
| | | | | | | 48.7 | 70.5 | 50.8 | 78.2 | 58.4 | 69.5 | 57.8 | 73.6 | 63.4 | - |
| ✓ | | | | | | 48.7 | 70.4 | 53.2 | 80.3 | 63.3 | 75.3 | 61.5 | 75.6 | 66.0 | +2.6 |
| ✓ | ✓ | | | | | 60.1 | 74.7 | 71.5 | 82.6 | 68.4 | 76.8 | 72.3 | 79.0 | 73.2 | +7.2 |
| ✓ | ✓ | ✓ | | | | 63.2 | 77.0 | 73.4 | 83.6 | 70.9 | 79.3 | 75.0 | 81.1 | 75.4 | +2.2 |
| ✓ | ✓ | ✓ | ✓ | | | 64.1 | 77.2 | 75.2 | 84.1 | 71.5 | 79.0 | 74.6 | 81.1 | **75.9** | +0.5 |
| ✓ | ✓ | ✓ | ✓ | ✓ | | 64.5 | 76.6 | 75.5 | 84.1 | 69.6 | 78.3 | 74.9 | 81.1 | 75.6 | −0.3 |
| ✓ | ✓ | ✓ | ✓ | | ✓ | 65.7 | 75.5 | 76.4 | 84.1 | 69.4 | 76.9 | 73.3 | 79.9 | 75.2 | −0.4 |
| ✓ | ✓ | ✓ | ✓ | ✓ | ✓ | 63.9 | 74.9 | 77.0 | 84.2 | 48.7 | 74.6 | 72.3 | 79.4 | 71.9 | −3.3 |

simultaneous application will over-emphasize tail classes and harm generalization. To balance these effects, we adopt Square-root sampling (a moderate re-sampling approach) and apply Balanced Softmax loss to the rectified distribution.

For our final framework, we integrate AutoAugment, Cosine classifier, Square-root sampling, Balanced Softmax loss, mixup, and label smoothing – all selected based on their excellent performance in previous experiments. We conduct ablation experiments on these methods under multiple settings, including different backbones such as CLIP and IN21K pre-trained ViT, and different fine-tuning methods such as full fine-tuning (FFT) and parameter-efficient fine-tuning (PEFT). The results are shown in Table 9. Due to the page limit, we report more detailed results for all datasets in Appendix section D.8.

From the results, we can conclude that 1) The combination methods of **Cosine Classifier**, **Square-root sampling**, **BS loss**, and **label smoothing** can **consistently** enhance the model performance on foundation models when using long-tailed data. As they achieve the best average performance across all scenarios, we consider the combination of these four methods as the optimal framework. 2) AutoAugment and mixup, as different forms of data augmentation, have **inconsistent** effects on performance across different datasets and models. There is no consistent conclusion on whether they improve or decrease performance based on our experiments, so we exclude them from the optimal framework.

Table 10: The results of applying our framework compared to other methods across four datasets: Places-LT, ImageNet-LT, CIFAR100-LT, iNaturalist 2018. † denotes VL-LTR uses extra data for fine-tuning. "-" means the paper has not reported the corresponding result. We also compare our framework with LIFT Shi et al. (2024) in Appendix D.9.

| | Places-LT | IN-LT | CIFAR-LT | iNat. |
| --- | --- | --- | --- | --- |
| MiSLAS (Zhong et al., 2021) | 40.4 | 52.7 | 47.0 | 71.6 |
| PaCo (Cui et al., 2021) | 41.2 | 57.0 | 52.0 | 71.8 |
| LiVT (Xu et al., 2023b) | 40.8 | 60.9 | 58.2 | 76.1 |
| BALLAD (Ma et al., 2021) | 49.5 | 75.7 | 77.8 | - |
| Decoder (Wang et al., 2024b) | 46.8 | 73.2 | - | 59.2 |
| LPT (Dong et al., 2023) | 50.1 | - | - | 76.1 |
| VL-LTR† (Tian et al., 2022) | 50.1 | **77.2** | - | 76.8 |
| Ours | **51.2** | **77.2** | **80.5** | **79.0** |

**Improvements over baselines** We apply our ultimate framework to four datasets on the pre-trained CLIP-ViT-B/16 backbone and obtain quite competitive results under PEFT settings. The test accuracy is reported in Table 10. Overall, our framework achieves superior performance on these challenging datasets, surpassing Decoder, LPT, VL-LTR, and various training-from-scratch approaches. And VL-LTR relies on extensive auxiliary data to facilitate fine-tuning, the advantage of our framework is more significant compared with methods that do not use auxiliary data. In ad-

dition, due to the Square-root sampling method included in our framework, the training cost of our framework is significantly reduced compared to other methods.

To provide a deeper mechanistic analysis. We examine the classifier weight norms, which led to some interesting findings. Specifically, we extract the classifier weight norms from models trained under the PEFT setting of CLIP-ViT/B-16 using four different datasets. Figure 4 displays the classifier weight norms for Cifar100-IR100 and Places-LT. The classes on the horizontal axis are arranged in descending order of their number of training samples.

Figure 4: Classifier weight norms for CIFAR100-IR100 and Places-LT.

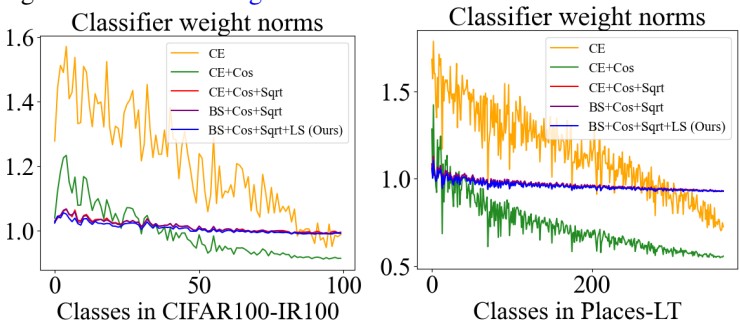

Due to the pronounced overlap between the curve of our method (blue line) and those of other approaches, visual inspection alone is insufficient to draw a definitive conclusion regarding its superior balance. To facilitate a quantitative comparison, we employ the standard deviation of the classifier weight norms as a metric for balance. The subsequent results are shown in Table 11 .

The results validate the superiority of our proposed method, which attains the most balanced norms — as evidenced by the lowest standard deviation in the comparison.

Table 11: Standard Deviation of classifier weight norms from models trained on different datasets. Each value in the table represents the actual standard deviation when multiplied by $10^{-2}$.

| Standard Deviation | Places-LT | IN-LT | CIFAR-LT | iNat. |
|---|---|---|---|---|
| CE | 16.3 | 23.8 | 13.6 | 10.6 |
| CE+Cos | 7.9 | 15.2 | 7.8 | 5.7 |
| CE+Cos+Sqrt | 1.8 | 3.0 | 1.3 | 3.9 |
| BS+Cos+Sqrt | 1.8 | 3.0 | 1.3 | 3.9 |
| BS+Cos+Sqrt+LS | **1.5** | **2.6** | **1.1** | **3.6** |

**Discussions**  We have taken into account the potential data leakage issue, such as between ImageNet and IN21K-ViT. In response to this, in Table 10, we only present results on CLIP-ViT-B/16. For detailed results across more experimental settings, we report in the Appendix. Looking ahead, we intend to explore the generalizability of our framework by extending it to more models, such as DINO (Oquab et al., 2023), which could further validate its transferability across different foundation models. Preliminary investigations in Appendix D.10 have already shown encouraging alignment with our current findings, suggesting broader applicability.

## 5 CONCLUSION

In this paper, we systematically revisit the representative long-tail learning methods and provide a scientific empirical guideline for their accurate use in fine-tuning foundation models. Furthermore, we select the optimal methods to construct a unified framework and analyze the contribution of each component through extensive ablation studies. Our proposed framework achieves competitive performance on multiple long-tailed datasets. We hope that our work serves as a convenient guideline for related applications and can inspire further research in the field of long-tail learning.

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

## A EXPERIMENTAL SETTINGS

### A.1 DATASETS

**CIFAR100-LT**  CIFAR100-LT is the long-tailed version of CIFAR (Krizhevsky, 2009). The latter is a balanced dataset consisting of 100 classes, with each class containing 500 samples for training and 100 samples for test. We construct CIFAR100-LT following the approach in (Cao et al., 2019). Specifically, each class contains $n_i = 500 \cdot r^{(-\frac{i-1}{99})}$ samples in training, where $i$ is class index. In this work, the imbalance factor is set to 100 considering its generality (Shi et al., 2024; Ma et al., 2021; Rangwani et al., 2022).

**Places-LT**  The Places-LT (Sharma et al., 2021) features a long-tailed dataset consisting of 62,500 images across 365 classes from Places-2 (Zhou et al., 2017). The class frequencies follow a natural power law distribution, with the largest class containing 4,980 images and the smallest class containing only 5 images.

**ImageNet-LT**  ImageNet-LT (Liu et al., 2019) is a long-tailed version of ImageNet ILSVRC 2012 (Deng et al., 2009), composed according to a Pareto distribution. This dataset consists of 1000 classes and a total of 1158K images, with the largest class containing up to 1,280 images and the smallest class containing as few as 5 images.

**iNaturalist 2018**  iNaturalist 2018 (Van Horn et al., 2018) is a natural dataset of fine-grained long-tailed categories, consisting of wildlife images across 8,142 species, with a total of 437,513 images. The number of images in each category ranges from a maximum of 1000 to a minimum of 2. It is a standard benchmark for evaluating algorithm performance on long-tailed distribution tasks.

### A.2 IMPLEMENTATION SETTINGS

In most of our experiments, we adopt pre-trained model CLIP (Radford et al., 2021) and Vision Transformer (Dosovitskiy et al., 2021) as the backbone and employ full fine-tuning (FFT) and parameter-efficient fine-tuning (PEFT) on these two models. Knowledge distillation is an exception where we use pre-trained DeiT (Touvron et al., 2021) as the student backbone. For the PEFT methods, we choose AdaptFormer (Chen et al., 2022) because of its optimal performance (Shi et al., 2024). Table 12 shows the performance of different PEFT methods under the ultimate framework. We use the SGD optimizer with a batch size of 128, weight decay of $5 \cdot 10^{-4}$, and momentum of 0.9. The number of training epochs for iNaturalist 2018 is 100, while for other datasets, it is 50. The learning rate is initialized to 0.1. The number of epochs and learning rate are carefully selected. We conduct comprehensive ablation studies on the epochs and learning rates across the CIFAR100-IR100, Places-LT, and ImageNet-LT datasets as shown in Table 13. We use mean accuracy and harmonic mean accuracy to measure the model's performance. In addition, we also follow the evaluation protocol introduced by (Liu et al., 2019), reporting accuracy for three categories: many-shot (>100 images), medium-shot (20-100 images), and few-shot (<20 images).

Table 12: Accuracy of using different PEFT methods.

| Datasets | Places-LT | | ImageNet-LT | |
|---|---|---|---|---|
| Backbone | CLIP | ViT | CLIP | ViT |
| LoRA | 50.7 | 47.1 | 76.0 | 83.8 |
| VPT-deep | 50.5 | 47.5 | 76.2 | **84.1** |
| Adapter | 50.9 | 47.7 | 77.0 | 84.0 |
| Bias-tuning | 50.9 | 47.3 | 76.2 | 83.2 |
| AdapterFormer | **51.2** | **47.9** | **77.2** | **84.1** |

## B KNOWLEDGE DISTILLATION

In this subsection, we focus on the knowledge distillation technique and explore whether it can improve the performance of long-tailed datasets on foundation models. We follow the setup mentioned in Data Efficient Transformer (DeiT) Touvron et al. (2021) to create the student backbone for our

Table 13: Comparison of different numbers of epochs and learning rate.

| | | CIFAR100-IR100 | Places-LT | ImageNet-LT |
|---|---|---|---|---|
| | 10 | 78.7 | 50.7 | 75.7 |
| | 20 | 80.2 | **51.3** | 77.0 |
| Epochs | 50 | **80.8** | 51.2 | **77.2** |
| | 70 | 80.7 | 50.8 | 77.0 |
| | 90 | 80.4 | 50.4 | 76.8 |
| | 0.001 | 78.7 | 49.8 | 74.4 |
| | 0.005 | **80.8** | **51.5** | 76.9 |
| LR | 0.01 | **80.8** | 51.2 | **77.2** |
| | 0.05 | 77.9 | 48.9 | 75.7 |
| | 0.1 | 77.1 | 48.3 | 75.3 |

experiments. In addition to the CLS token, DeiT adds a DIST token in the ViT backbone that learns via distillation from the teacher. For both the classification head and the distillation head, training is conducted using cross-entropy loss, and the final loss function Rangwani et al. (2024) is

$$\mathcal{L} = aL_{CE}(f^{cls}(x), y) + (1-a)L_{CE}(f^{dis}(x), y_t) \tag{4}$$

where $f^{cls}(x)$ and $f^{dis}(x)$ are outputs of the CLS and DIST tokens through their respective layers, $y$ is the ground truth, and $y_t$ is the teacher model's hard label for sample $x$.

**Experimental Result**  We simply set the $a$ to $0.5$ to ensure the fair status of the ground truth and the teacher's prediction. Table 14 shows the accuracy of the knowledge distillation methods. For more detailed settings and results, please refer to Appendix section D.5.

Compared to PEFT, the performance enhancement under FFT is significantly more substantial. Experimental results demonstrate that knowledge distillation yields an improvement of approximately 3% in the FFT setting, whereas it contributes almost no gain in the PEFT setting.

We believe this is because knowledge distillation helps mitigate the biases towards the head classes in the student model during training. Since the FFT setting involves substantially more parameters to train compared to the PEFT setting, it is more susceptible to being biased toward head classes. This explains why the performance improvements are more pronounced in the FFT setting.

Table 14: Student results of applying knowledge distillation.

| Datasets | CIFAR100-LT | | | | Places-LT | | | |
|---|---|---|---|---|---|---|---|---|
| Student | DeiT-S | | DeiT-Ti | | DeiT-S | | DeiT-Ti | |
| | FFT | PEFT | FFT | PEFT | FFT | PEFT | FFT | PEFT |
| Baseline | 67.3 | 69.9 | 58.7 | **60.8** | 27.1 | 32.1 | 24.6 | 29.4 |
| Distillation | **70.4** | **70.0** | **61.7** | 60.6 | **30.2** | **32.5** | **28.6** | **30.0** |

## C  ENSEMBLE LEARNING

Ensemble learning improves model performance by combining the predictions of multiple experts to address the long-tail problem. We conduct an experiment using a framework similar to BBN Zhou et al. (2020). Specifically, we use two branches: the "conventional learning branch", which employs the uniform sampler to learn the original data distribution, and the "re-balancing branch", which uses the reversed sampler to sample more tail-class samples for learning a balanced distribution. Both branches use the same backbone and share all the weights except for the last classifier. At last, a cumulative loss weight $w$ is used to shift the learning "attention" smoothly from the head class to the tail class. Formally, the objective loss of the model is illustrated as

$$\mathcal{L} = wL_{CE}(f^c(x^c), y^c) + (1-w)L_{CE}(f^r(x^r), y^r) \tag{5}$$

$$w = 1 - (\frac{t_c}{t_{max}})^2 \tag{6}$$

where the $f^c(x^c)$ and $f^r(x^r)$ respectively represent the predicted output of the conventional learning branch and re-balancing branch. $y^c$ and $y^r$ are the ground truth of $x^c$ and $x^r$ respectively. $t_c$ and $t_{max}$ respectively refer to the current epoch and total training epochs.

**Experimental Result** Ensemble-based methods address the class imbalance at the model level. Table 15 shows the accuracy of the ensemble method. For more detailed results, please refer to Appendix section D.6. Ensemble methods can generally improve performance by an average of over 3% in the PEFT setting. However, in the FFT setting, the model improvements are less favorable, with a maximum increase of 1%, and in some cases, even face a significant decrease.

Additionally, it is very important to note that ensemble learning inevitably increases the training cost. In this experiment, using two branches **doubles** the memory cost and computational time expenditure, because we need to create two individual data samplers and calculate the corresponding loss. In practice, though more experts may lead to better performance, the greater time and storage costs are non-negligible overheads. Therefore, we only recommend employing ensemble learning in the lightweight PEFT setting on foundation models. Using ensemble learning in the FFT setting is not cost-effective and does not guarantee performance improvements.

Table 15: Accuracy of applying ensemble learning.

| Datasets | CIFAR100-LT | | | | Places-LT | | | |
|---|---|---|---|---|---|---|---|---|
| Backbone | CLIP | | ViT | | CLIP | | ViT | |
| | FFT | PEFT | FFT | PEFT | FFT | PEFT | FFT | PEFT |
| Baseline | 54.6 | 71.9 | **70.3** | 80.7 | **24.9** | 39.8 | 26.0 | 31.9 |
| Ensemble | **55.6** | **76.0** | 68.6 | **82.2** | 18.9 | **45.0** | **26.7** | **36.4** |

# D    ADDITIONAL RESULTS

## D.1    RE-SAMPLING DETAILED RESULTS

For re-sampling methods, we report detailed results of applying RUS, RUSxN, ROS, EQ, Square-root sampling and no resampling (Baseline) methods. Tables 16, 17, 18 show the detailed results of applying re-sampling methods for CIFAR100-LT. Places-LT, ImageNet-LT respectively. Tables 19, 20, 21 show the detailed results of applying RUSxN for CIFAR100-LT, Places-LT and ImageNet-LT respectively. We can observe that applying RUS and Square-root sampling can significantly improve model performance.

## D.2    DATA AUGMENTATION DETAILED RESULTS

For data augmentation methods, we report detailed results of applying ColorJitter, RandAugment, AutoAugment, and no augmentation (Baseline) methods. Tables 22, 23, 24 show the detailed results of applying data augmentation methods for CIFAR100-LT, Places-LT, ImageNet-LT respectively. We can observe that applying data augmentation methods can only slightly improve the model performance and don't play a decisive role.

## D.3    CLASS-SENSITIVE LOSS DETAILED RESULTS

For Class-sensitive loss, we report detailed results of applying CE, Focal, Label-Distribution-Aware Margin, Class-Balanced, Generalized Re-Weight, Balanced Softmax, Logit Adjustment, LAbel distribution DisEntangling loss. The selection of hyperparameters for each loss follows the corresponding paper, except for G-RW. The original paper of G-RW proposed $\rho = 1.2$, which performs very poorly under FFT settings for each backbone. After our experimental attempts, we finally changed it to $0.5$. The selected hyperparameters are shown as follows:

Focal loss: $\gamma = 2$; LDAM loss: $s = 25$; Class Balanced loss: $\beta = 0.9$; Generalized Re-weight loss: $\rho = 0.5$ for FFT setting, $\rho = 1.2$ for PEFT setting; Logit adjustment loss: $\mu = 1.5$; LADE loss: $\alpha = 0.01, \lambda = 0.1$.

In practice, we have tried different hyperparameters but only report the best. For example, we have tried: $\gamma = \{2, 3, 4\}$ for Focal loss; $\beta = \{0.9, 0.99, 0.999\}$ for Class-Balanced loss; $\tau = \{1, 1.5, 2\}$ for LA loss; $\rho = \{0.5, 1, 1.2, 1.5, 2\}$ for G-RW loss.

Tables 25, 26, 27 show the detailed results of applying class-sensitive losses for CIFAR100-LT, Places-LT, ImageNet-LT respectively. We can observe that applying Balanced Softmax loss and Logit Adjustment loss can greatly gain benefits.

### D.4 BALANCED CLASSIFIER DETAILED RESULTS

For the balanced classifier, we report detailed results of using the Cosine classifier, $\tau$-normalized classifier, and Linear classifier methods. Tables 28, 29, 30 show the detailed results of applying different classifiers for CIFAR100-LT, Places-LT, ImageNet-LT respectively. We can observe that Cosine classifier can achieve an improvement in model performance.

### D.5 KNOWLEDGE DISTILLATION DETAILED RESULTS

We use a well-trained CLIP-ViT-B/16 as the teacher backbone for Places-LT and IN21K-ViT-B/16 as the teacher backbone for CIFAR100-LT and ImageNet-LT, while employing the pre-trained DeiT-S and DeiT-Ti backbone architecture as student models for all the datasets. Tables 31, 32, 33 show the detailed results of applying knowledge distillation on CIFAR100-LT, Places-LT, ImageNet-LT respectively. We can observe that knowledge distillation is only effective in the FFT setting.

### D.6 ENSEMBLE LEARNING DETAILED RESULTS

We build a framework similar to BBN and report details results of applying it on CIFAR100-LT, Places-LT and ImageNet-LT as shown in Tables 34, 35, 36 respectively. We can observe that applying ensemble learning is only cost-effective under the PEFT setting.

### D.7 TRICKS DETAILED RESULTS

For tricks, we report detailed results of applying mixup and label smoothing. Tables 37, 38, 39 show the detailed results of applying mixup for CIFAR100-LT, Places-LT, ImageNet-LT respectively. Tables 40, 41, 42 show the detailed results of applying label smoothing for CIFAR100-LT, Places-LT, ImageNet-LT respectively. We can observe that both tricks can improve model performance.

### D.8 ABLATION EXPERIMENTS DETAILED RESULTS

To build the best framework for fine-tuning pre-trained models, we choose AutoAugment, Cosine classifier, Square-root resampling, Balanced Softmax loss, Mixup, and Label smoothing for the ablation experiments.

Tables 43, 44, 45, 46 show the detailed ablation results for CIFAR100-LT, Places-LT, ImageNet-LT, iNaturalist 2018 datasets respectively.

### D.9 COMPARISON WITH LIFT

The performance of our model is comparable to that achieved by LIFT, as shown in the tables 47. Although we have more epochs, due to the sampling strategy of the data, the total training cost is significantly lower compared to LIFT, **with an average saved cost of 21%** (specific values vary depending on the dataset). Notably, it achieves a remarkable 34% reduction on the Places-LT, demonstrating the effectiveness of our method.

### D.10 TRANSFERABILITY OF OUR FRAMEWORK

To verify the transferability of our framework, we extend it to DINO and conduct corresponding experiments. The results are shown in Table 48. we are temporarily unable to report results for DINOv2 due to GPU memory limitations. Our framework can also be readily adapted to MAE and SigLIP, which are planned for a future version.

Table 16: Detailed results of applying resampling methods to the CIFAR100-LT dataset.

| | | | Mean | Many | Med. | Few | Harmonic mean | Worst case |
|---|---|---|---|---|---|---|---|---|
| CLIP-ViT-B/16 | FFT | Baseline | 54.6 | 82.9 | 55.9 | 20.1 | 0.0 | 0.0 |
| | | Random Over-Sampling | 44.8 | 77.7 | 42.5 | 9.1 | 0.0 | 0.0 |
| | | Random Under-Sampling | 45.5 | 51.0 | 49.8 | 34.1 | 31.0 | 6.0 |
| | | Equal resampling | 50.4 | 82.6 | 50.5 | 12.8 | 0.0 | 0.0 |
| | | Square-root resampling | 56.8 | 80.3 | 60.0 | 25.8 | 0.1 | 0.0 |
| | PEFT | Baseline | 71.9 | 90.2 | 75.1 | 46.6 | 56.0 | 7.0 |
| | | Random Over-Sampling | 68.3 | 89.0 | 73.1 | 38.5 | 36.9 | 2.0 |
| | | Random Under-Sampling | 77.4 | 79.9 | 79.1 | 72.5 | 73.6 | 26.0 |
| | | Equal resampling | 72.8 | 88.6 | 77.2 | 49.2 | 55.9 | 7.0 |
| | | Square-root resampling | 76.4 | 87.1 | 78.2 | 61.8 | 69.8 | 18.0 |
| IN21K-ViT-B/16 | FFT | Baseline | 70.3 | 89.6 | 71.9 | 45.8 | 48.0 | 3.0 |
| | | Random Over-Sampling | 48.3 | 83.4 | 46.0 | 10.0 | 0.0 | 0.0 |
| | | Random Under-Sampling | 69.3 | 74.6 | 71.6 | 60.3 | 58.5 | 5.0 |
| | | Equal resampling | 62.0 | 90.0 | 64.6 | 26.2 | 0.0 | 0.0 |
| | | Square-root resampling | 76.0 | 90.5 | 78.1 | 56.6 | 60.2 | 5.0 |
| | PEFT | Baseline | 80.7 | 93.5 | 80.9 | 65.4 | 41.2 | 1.0 |
| | | Random Over-Sampling | 71.0 | 93.0 | 76.3 | 39.2 | 0.1 | 0.0 |
| | | Random Under-Sampling | 87.0 | 90.5 | 87.3 | 82.6 | 81.9 | 15.0 |
| | | Equal resampling | 77.3 | 93.1 | 80.3 | 55.3 | 37.3 | 1.0 |
| | | Square-root resampling | 84.4 | 93.8 | 85.1 | 72.6 | 69.2 | 7.0 |

Table 17: Detailed results of applying resampling methods to the Places-LT dataset.

| | | | Mean | Many | Med. | Few | Harmonic mean | Worst case |
|---|---|---|---|---|---|---|---|---|
| CLIP-ViT-B/16 | FFT | Baseline | 24.7 | 40.5 | 19.8 | 6.8 | 0.1 | 0.0 |
| | | Random Over-Sampling | 12.6 | 25.7 | 7.2 | 0.7 | 0.0 | 0.0 |
| | | Random Under-Sampling | 42.3 | 42.6 | 45.4 | 34.5 | 0.2 | 0.0 |
| | | Equal resampling | 21.7 | 40.2 | 14.7 | 3.6 | 0.0 | 0.0 |
| | | Square-root resampling | 37.1 | 51.0 | 34.8 | 16.6 | 18.8 | 1.0 |
| | PEFT | Baseline | 39.8 | 54.0 | 35.7 | 22.7 | 0.1 | 0.0 |
| | | Random Over-Sampling | 38.3 | 51.0 | 35.5 | 20.9 | 0.4 | 0.0 |
| | | Random Under-Sampling | 50.8 | 49.6 | 52.2 | 49.6 | 35.7 | 1.0 |
| | | Equal resampling | 43.7 | 53.3 | 42.8 | 27.9 | 25.2 | 1.0 |
| | | Square-root resampling | 47.5 | 55.6 | 45.7 | 36.6 | 32.4 | 2.0 |
| IN21K-ViT-B/16 | FFT | Baseline | 26.0 | 41.4 | 20.9 | 9.5 | 0.1 | 0.0 |
| | | Random Over-Sampling | 11.4 | 24.2 | 5.7 | 0.7 | 0.0 | 0.0 |
| | | Random Under-Sampling | 41.2 | 47.6 | 43.3 | 24.6 | 23.4 | 1.0 |
| | | Equal resampling | 22.0 | 40.4 | 14.9 | 4.2 | 0.0 | 0.0 |
| | | Square-root resampling | 32.6 | 48.6 | 27.7 | 14.1 | 0.2 | 0.0 |
| | PEFT | Baseline | 32.1 | 45.9 | 28.4 | 15.2 | 0.2 | 0.0 |
| | | Random Over-Sampling | 32.2 | 45.8 | 28.9 | 14.9 | 0.1 | 0.0 |
| | | Random Under-Sampling | 45.3 | 46.7 | 47.6 | 37.5 | 32.2 | 2.0 |
| | | Equal resampling | 33.7 | 47.5 | 30.7 | 15.2 | 0.1 | 0.0 |
| | | Square-root resampling | 39.7 | 50.9 | 37.7 | 23.3 | 23.5 | 2.0 |

Table 18: Detailed results of applying resampling methods to the ImageNet-LT dataset."-" means the corresponding experiment is hard to implement due to the huge amount of data.

| | | | Mean | Many | Med. | Few | Harmonic mean | Worst case |
|---|---|---|---|---|---|---|---|---|
| CLIP-ViT-B/16 | FFT | Baseline | 49.9 | 69.0 | 44.0 | 16.6 | 0.0 | 0.0 |
| | | Random Over-Sampling | - | - | - | - | - | - |
| | | Random Under-Sampling | 59.2 | 62.1 | 59.0 | 52.2 | 44.6 | 2.0 |
| | | Equal resampling | 48.6 | 67.8 | 41.9 | 18.0 | 0.0 | 0.0 |
| | | Square-root resampling | 59.9 | 74.8 | 56.1 | 31.2 | 0.2 | 0.0 |
| | PEFT | Baseline | 70.6 | 85.5 | 67.6 | 38.8 | 0.1 | 0.0 |
| | | Random Over-Sampling | - | - | - | - | - | - |
| | | Random Under-Sampling | 75.4 | 78.2 | 75 | 68.4 | 67.6 | 10.0 |
| | | Equal resampling | 73.6 | 83.2 | 72.2 | 51.1 | 1.0 | 0.0 |
| | | Square-root resampling | 74.5 | 83.9 | 72.5 | 54.7 | 59.7 | 2.0 |
| IN21K-ViT-B/16 | FFT | Baseline | 52.1 | 70.1 | 45.9 | 23.0 | 0.1 | 0.0 |
| | | Random Over-Sampling | - | - | - | - | - | - |
| | | Random Under-Sampling | 72.6 | 79.2 | 71.7 | 57.0 | 1.0 | 0.0 |
| | | Equal resampling | 50.1 | 70.1 | 43.1 | 18.7 | 0.0 | 0.0 |
| | | Square-root resampling | 68.2 | 80.6 | 64.8 | 44.8 | 1.0 | 0.0 |
| | PEFT | Baseline | 78.2 | 87.5 | 75.8 | 59.9 | 64.4 | 2.0 |
| | | Random Over-Sampling | - | - | - | - | - | - |
| | | Random Under-Sampling | 83.2 | 85.6 | 82.9 | 77.4 | 78.9 | 16.0 |
| | | Equal resampling | 79.2 | 87.4 | 77.4 | 61.8 | 69.7 | 8.0 |
| | | Square-root resampling | 81.0 | 87.3 | 79.5 | 68.6 | 74.1 | 8.0 |

Table 19: Detailed results of applying RUSxN to the CIFAR100-LT dataset.

| | | | Mean | Many | Med. | Few | Harmonic mean | Worst case |
|---|---|---|---|---|---|---|---|---|
| CLIP-ViT-B/16 | FFT | RUS | 56.0 | 71.6 | 61.3 | 31.5 | 21.8 | 1.0 |
| | | RUSx2 | 58.0 | 81.3 | 62.3 | 25.8 | 0.0 | 0.0 |
| | | RUSx5 | 54.0 | 83.0 | 55.5 | 18.3 | 0.0 | 0.0 |
| | | RUSx10 | 48.8 | 79.1 | 49.5 | 12.8 | 0.0 | 0.0 |
| | PEFT | RUS | 77.7 | 82.0 | 80.0 | 69.9 | 73.7 | 25.0 |
| | | RUSx2 | 77.5 | 85.3 | 80.6 | 64.6 | 71.6 | 20.0 |
| | | RUSx5 | 75.6 | 87.2 | 79.7 | 57.4 | 63.4 | 8.0 |
| | | RUSx10 | 73.4 | 88.1 | 77.8 | 50.7 | 56.8 | 6.0 |
| IN21K-ViT-B/16 | FFT | RUS | 75.7 | 87.9 | 78.1 | 58.6 | 59.6 | 5.0 |
| | | RUSx2 | 70.8 | 90.6 | 73.1 | 44.9 | 35.2 | 1.0 |
| | | RUSx5 | 66.4 | 90.6 | 69.5 | 34.4 | 26.5 | 1.0 |
| | | RUSx10 | 59.8 | 87.7 | 62.1 | 24.4 | 0.1 | 0.0 |
| | PEFT | RUS | 86.3 | 91.4 | 87.5 | 79.2 | 77.6 | 11.0 |
| | | RUSx2 | 84.5 | 92.7 | 86.4 | 72.8 | 71.0 | 8.0 |
| | | RUSx5 | 81.0 | 93.5 | 82.7 | 64.5 | 41.6 | 1.0 |
| | | RUSx10 | 78.2 | 93.4 | 80.2 | 58.3 | 39.5 | 1.0 |

Table 20: Detailed results of applying RUSxN to the Places-LT dataset.

| | | | Mean | Many | Med. | Few | Harmonic mean | Worst case |
|---|---|---|---|---|---|---|---|---|
| CLIP-ViT-B/16 | FFT | RUS | 42.3 | 42.6 | 45.4 | 34.5 | 0.2 | 0.0 |
| | | RUSx2 | 41.6 | 46.6 | 45.3 | 24.1 | 24.8 | 1.0 |
| | | RUSx5 | 34.6 | 49.4 | 32.9 | 10.9 | 0.1 | 0.0 |
| | | RUSx10 | 29.0 | 48.3 | 23.3 | 6.6 | 0.0 | 0.0 |
| | PEFT | RUS | 50.8 | 49.6 | 52.2 | 49.6 | 35.7 | 1.0 |
| | | RUSx2 | 50.6 | 50.5 | 52.6 | 46.1 | 35.8 | 1.0 |
| | | RUSx5 | 49.1 | 51.5 | 51.4 | 39.2 | 35.5 | 3.0 |
| | | RUSx10 | 47.7 | 52.7 | 48.9 | 33.9 | 0.4 | 0.0 |
| IN21K-ViT-B/16 | FFT | RUS | 41.2 | 47.6 | 43.3 | 24.6 | 23.4 | 1.0 |
| | | RUSx2 | 38.0 | 50.7 | 36.9 | 16.8 | 0.1 | 0.0 |
| | | RUSx5 | 31.6 | 49.4 | 26.3 | 10.8 | 0.1 | 0.0 |
| | | RUSx10 | 27.7 | 46.3 | 21.2 | 8.4 | 0.1 | 0.0 |
| | PEFT | RUS | 45.3 | 46.7 | 47.6 | 37.5 | 32.2 | 2.0 |
| | | RUSx2 | 43.2 | 48.4 | 44.9 | 29.9 | 29.4 | 3.0 |
| | | RUSx5 | 39.1 | 49.2 | 38.5 | 21.6 | 0.2 | 0.0 |
| | | RUSx10 | 29.0 | 48.3 | 23.3 | 6.6 | 0.0 | 0.0 |

Table 21: Detailed results of applying RUSxN to the ImageNet-LT dataset.

| | | | Mean | Many | Med. | Few | Harmonic mean | Worst case |
|---|---|---|---|---|---|---|---|---|
| CLIP-ViT-B/16 | FFT | RUS | 59.2 | 62.1 | 59.0 | 52.2 | 44.6 | 2.0 |
| | | RUSx2 | 61.3 | 68.5 | 61.2 | 41.4 | 1.0 | 0.0 |
| | | RUSx5 | 59.0 | 72.8 | 56.0 | 30.6 | 0.2 | 0.0 |
| | | RUSx10 | 55.2 | 72.2 | 50.2 | 24.3 | 0.1 | 0.0 |
| | PEFT | RUS | 75.4 | 78.2 | 75.0 | 68.4 | 67.6 | 10.0 |
| | | RUSx2 | 75.9 | 80.0 | 75.5 | 65.9 | 67.9 | 6.0 |
| | | RUSx5 | 75.7 | 81.5 | 75.4 | 60.3 | 66.6 | 8.0 |
| | | RUSx10 | 75.0 | 82.4 | 74.3 | 56.0 | 62.5 | 4.0 |
| IN21K-ViT-B/16 | FFT | RUS | 72.6 | 79.2 | 71.7 | 57.0 | 1.0 | 0.0 |
| | | RUSx2 | 71.5 | 80.9 | 69.6 | 51.5 | 58.2 | 2.0 |
| | | RUSx5 | 66.0 | 79.9 | 61.9 | 40.9 | 1.0 | 0.0 |
| | | RUSx10 | 60.5 | 77.2 | 55.7 | 30.7 | 0.2 | 0.0 |
| | PEFT | RUS | 83.2 | 85.6 | 82.9 | 77.4 | 78.9 | 16.0 |
| | | RUSx2 | 82.7 | 86.0 | 82.3 | 74.5 | 78.3 | 18.0 |
| | | RUSx5 | 80.6 | 86.7 | 79.3 | 67.8 | 73.7 | 10.0 |
| | | RUSx10 | 79.3 | 87.0 | 77.6 | 63.7 | 70.6 | 8.0 |

Table 22: Detailed results of applying augmentation methods to the CIFAR100-LT dataset.

| | | | Mean | Many | Med. | Few | Harmonic mean | Worst case |
|---|---|---|---|---|---|---|---|---|
| CLIP-ViT-B/16 | FFT | Baseline | 48.7 | 77.9 | 48.1 | 15.4 | 12.2 | 1.0 |
| | | ColorJitter | 55.0 | 83.2 | 56.3 | 20.7 | 0.1 | 0.0 |
| | | RandAugment | 56.7 | 84.1 | 57.9 | 23.5 | 0.1 | 0.0 |
| | | AutoAugment | 57.8 | 85.5 | 58.5 | 24.7 | 0.1 | 0.0 |
| | PEFT | Baseline | 71.9 | 90.0 | 75.3 | 46.9 | 57.6 | 9.0 |
| | | ColorJitter | 71.9 | 90.2 | 75.1 | 46.6 | 56.0 | 7.0 |
| | | RandAugment | 72.1 | 90.1 | 75.4 | 47.3 | 54.7 | 7.0 |
| | | AutoAugment | 70.1 | 90.1 | 73.8 | 44.5 | 36.1 | 1.0 |
| IN21K-ViT-B/16 | FFT | Baseline | 71.1 | 89.3 | 72.6 | 48.0 | 50.6 | 3.0 |
| | | ColorJitter | 70.3 | 89.6 | 71.9 | 45.8 | 48.0 | 3.0 |
| | | RandAugment | 70.0 | 89.6 | 70.7 | 46.3 | 45.4 | 2.0 |
| | | AutoAugment | 71.6 | 90.7 | 72.3 | 48.4 | 54.2 | 7.0 |
| | PEFT | Baseline | 81.6 | 93.3 | 81.9 | 67.6 | 41.9 | 1.0 |
| | | ColorJitter | 80.7 | 93.5 | 80.9 | 65.4 | 41.2 | 1.0 |
| | | RandAugment | 81.5 | 93.7 | 81.5 | 67.2 | 54.7 | 3.0 |
| | | AutoAugment | 81.3 | 93.3 | 81.8 | 66.7 | 42.2 | 1.0 |

Table 23: Detailed results of applying augmentation methods to the Places-LT dataset.

| | | | Mean | Many | Med. | Few | Harmonic mean | Worst case |
|---|---|---|---|---|---|---|---|---|
| CLIP-ViT-B/16 | FFT | Baseline | 23.7 | 39.8 | 18.5 | 6.0 | 0.1 | 0.0 |
| | | ColorJitter | 24.7 | 40.5 | 19.8 | 6.8 | 0.1 | 0.0 |
| | | RandAugment | 25.4 | 41.6 | 20.4 | 6.9 | 0.1 | 0.0 |
| | | AutoAugment | 24.9 | 41.8 | 19.8 | 5.6 | 0.0 | 0.0 |
| | PEFT | Baseline | 39.8 | 54.5 | 35.7 | 22.3 | 0.1 | 0.0 |
| | | ColorJitter | 39.8 | 54.0 | 35.7 | 22.7 | 0.1 | 0.0 |
| | | RandAugment | 40.4 | 54.7 | 36.5 | 23.1 | 0.1 | 0.0 |
| | | AutoAugment | 40.7 | 54.9 | 36.8 | 23.2 | 0.1 | 0.0 |
| IN21K-ViT-B/16 | FFT | Baseline | 25.7 | 40.9 | 20.5 | 9.4 | 0.1 | 0.0 |
| | | ColorJitter | 26.0 | 41.4 | 20.9 | 9.5 | 0.1 | 0.0 |
| | | RandAugment | 26.5 | 41.9 | 21.6 | 9.4 | 0.1 | 0.0 |
| | | AutoAugment | 26.9 | 42.1 | 22.1 | 9.7 | 0.1 | 0.0 |
| | PEFT | Baseline | 31.7 | 45.5 | 27.8 | 15.1 | 0.1 | 0.0 |
| | | ColorJitter | 32.1 | 45.9 | 28.4 | 15.2 | 0.2 | 0.0 |
| | | RandAugment | 32.6 | 46.8 | 28.8 | 15.4 | 0.2 | 0.0 |
| | | AutoAugment | 32.7 | 46.8 | 29.1 | 15.2 | 0.2 | 0.0 |

Table 24: Detailed results of applying augmentation methods to the ImageNet-LT dataset.

| | | | Mean | Many | Med. | Few | Harmonic mean | Worst case |
|---|---|---|---|---|---|---|---|---|
| CLIP-ViT-B/16 | FFT | Baseline | 48.7 | 67.9 | 42.5 | 16.0 | 0.0 | 0.0 |
| | | ColorJitter | 49.9 | 69.0 | 44.0 | 16.6 | 0.0 | 0.0 |
| | | RandAugment | 51.0 | 70.2 | 45.1 | 17.6 | 0.0 | 0.0 |
| | | AutoAugment | 51.8 | 71.3 | 45.9 | 17.0 | 0.0 | 0.0 |
| | PEFT | Baseline | 70.5 | 85.5 | 67.5 | 38.3 | 0.1 | 0.0 |
| | | ColorJitter | 70.6 | 85.5 | 67.6 | 38.8 | 0.1 | 0.0 |
| | | RandAugment | 70.5 | 85.5 | 67.5 | 38.3 | 0.1 | 0.0 |
| | | AutoAugment | 70.3 | 81.0 | 67.2 | 38.2 | 0.1 | 0.0 |
| IN21K-ViT-B/16 | FFT | Baseline | 50.8 | 69.1 | 44.4 | 21.8 | 0.1 | 0.0 |
| | | ColorJitter | 52.1 | 70.1 | 45.9 | 23.0 | 0.1 | 0.0 |
| | | RandAugment | 53.4 | 71.4 | 47.1 | 24.5 | 0.1 | 0.0 |
| | | AutoAugment | 54.1 | 72.1 | 48.1 | 24.2 | 0.1 | 0.0 |
| | PEFT | Baseline | 78.2 | 87.4 | 76.0 | 59.8 | 1.0 | 0.0 |
| | | ColorJitter | 78.2 | 87.5 | 75.8 | 59.9 | 64.4 | 2.0 |
| | | RandAugment | 78.1 | 87.5 | 75.7 | 59.6 | 63.8 | 2.0 |
| | | AutoAugment | 78.2 | 87.5 | 75.9 | 60.1 | 66.2 | 6.0 |

Table 25: Detailed results of applying different losses to the CIFAR100-LT dataset.

| | | | Mean | Many | Med. | Few | Harmonic mean | Worst case |
|---|---|---|---|---|---|---|---|---|
| CLIP-ViT-B/16 | FFT | CE loss | 54.6 | 82.9 | 55.9 | 20.1 | 0.0 | 0.0 |
| | | Focal loss | 52.7 | 81.4 | 53.0 | 19.0 | 17.9 | 1.0 |
| | | LDAM loss | 53.6 | 78.5 | 52.9 | 25.4 | 0.1 | 0.0 |
| | | Class Balanced loss | 54.7 | 83.4 | 56.1 | 19.4 | 0.1 | 0.0 |
| | | Generalized Re-Weight | 50.9 | 80.4 | 50.9 | 16.5 | 0.0 | 0.0 |
| | | Balanced Softmax Loss | 58.0 | 75.5 | 58.9 | 36.5 | 42.3 | 6.0 |
| | | Logit Adjustment loss | 62.7 | 74.8 | 62.1 | 49.4 | 55.3 | 18.0 |
| | | LADE loss | 18.2 | 26.3 | 19.9 | 6.9 | 0.0 | 0.0 |
| | PEFT | CE loss | 71.9 | 90.2 | 75.1 | 46.6 | 56.0 | 7.0 |
| | | Focal loss | 71.2 | 89.5 | 74.0 | 46.7 | 58.3 | 10.0 |
| | | LDAM loss | 73.6 | 89.5 | 77.4 | 50.7 | 0.1 | 0.0 |
| | | Class Balanced loss | 72.5 | 90.2 | 75.3 | 48.6 | 57.7 | 9.0 |
| | | Generalized Re-Weight | 71.8 | 84.0 | 78.3 | 50.1 | 55.8 | 9.0 |
| | | Balanced Softmax Loss | 80.1 | 86.5 | 80.0 | 72.9 | 77.5 | 38.0 |
| | | Logit Adjustment loss | 79.8 | 80.6 | 79.3 | 79.5 | 77.8 | 47.0 |
| | | LADE loss | 79.9 | 85.7 | 79.0 | 74.1 | 77.1 | 42.0 |
| IN21K-ViT-B/16 | FFT | CE loss | 70.3 | 89.6 | 71.9 | 45.8 | 48.0 | 3.0 |
| | | Focal loss | 69.4 | 89.3 | 71.1 | 44.3 | 43.9 | 2.0 |
| | | LDAM loss | 64.4 | 85.9 | 67.0 | 36.3 | 41.5 | 4.0 |
| | | Class Balanced loss | 69.4 | 89.6 | 71.3 | 43.7 | 50.1 | 5.0 |
| | | Generalized Re-Weight | 66.9 | 88.9 | 69.5 | 38.1 | 0.0 | 0.1 |
| | | Balanced Softmax Loss | 75.8 | 88.7 | 76.4 | 59.9 | 63.0 | 6.0 |
| | | Logit Adjustment loss | 73.1 | 88.4 | 73.1 | 55.1 | 64.7 | 15.0 |
| | | LADE loss | 72.8 | 89.9 | 72.4 | 53.2 | 48.8 | 2.0 |
| | PEFT | CE loss | 80.7 | 93.5 | 80.9 | 65.4 | 41.2 | 1.0 |
| | | Focal loss | 81.4 | 93.5 | 81.3 | 67.5 | 52.9 | 2.0 |
| | | LDAM loss | 82.8 | 93.3 | 83.3 | 70.0 | 67.6 | 6.0 |
| | | Class Balanced loss | 80.3 | 93.4 | 80.7 | 64.6 | 41.7 | 1.0 |
| | | Generalized Re-Weight | 81.8 | 93.0 | 84.0 | 66.3 | 62.9 | 5.0 |
| | | Balanced Softmax Loss | 85.1 | 92.0 | 84.8 | 77.4 | 79.5 | 18.0 |
| | | Logit Adjustment loss | 86.3 | 91.9 | 85.9 | 81.3 | 83.2 | 28.0 |
| | | LADE loss | 86.0 | 93.0 | 85.0 | 79.2 | 81.6 | 23.0 |

Table 26: Detailed results of applying different losses to the Places-LT dataset.

| | | | Mean | Many | Med. | Few | Harmonic mean | Worst case |
|---|---|---|---|---|---|---|---|---|
| CLIP-ViT-B/16 | FFT | CE loss | 24.7 | 40.5 | 19.8 | 6.8 | 0.1 | 0.0 |
| | | Focal loss | 24.3 | 40.5 | 19.2 | 6.3 | 0.1 | 0.0 |
| | | LDAM loss | 24.7 | 37.9 | 21.2 | 8.6 | 0.0 | 0.0 |
| | | Class Balanced loss | 25.1 | 40.7 | 20.3 | 7.3 | 0.0 | 0.0 |
| | | Generalized Re-Weight | 22.0 | 38.8 | 16.3 | 4.1 | 0.0 | 0.0 |
| | | Balanced Softmax Loss | 31.3 | 39.7 | 28.0 | 23.3 | 20.6 | 3.0 |
| | | Logit Adjustment loss | 32.0 | 36.2 | 29.9 | 29.3 | 21.6 | 3.0 |
| | | LADE loss | 16.8 | 23.0 | 16.7 | 5.5 | 0.0 | 0.0 |
| | PEFT | CE loss | 39.8 | 53.9 | 35.9 | 22.5 | 0.1 | 0.0 |
| | | Focal loss | 39.0 | 52.9 | 35.1 | 22.1 | 0.2 | 0.0 |
| | | LDAM loss | 41.1 | 54.7 | 37.4 | 24.3 | 0.0 | 0.0 |
| | | Class Balanced loss | 40.2 | 54.0 | 35.8 | 24.6 | 0.1 | 0.0 |
| | | Generalized Re-Weight | 44.5 | 51.1 | 46.3 | 28.2 | 0.4 | 0.0 |
| | | Balanced Softmax Loss | 48.8 | 49.7 | 49.0 | 46.9 | 39.4 | 4.0 |
| | | Logit Adjustment loss | 48.0 | 41.4 | 50.5 | 54.7 | 0.4 | 0.0 |
| | | LADE loss | 49.2 | 49.9 | 49.3 | 47.6 | 35.4 | 1.0 |
| IN21K-ViT-B/16 | FFT | CE loss | 26.0 | 41.4 | 20.9 | 9.5 | 0.1 | 0.0 |
| | | Focal loss | 25.9 | 41.2 | 21.0 | 8.8 | 0.1 | 0.0 |
| | | LDAM loss | 25.0 | 39.9 | 20.0 | 9.1 | 0.1 | 0.0 |
| | | Class Balanced loss | 26.0 | 41.2 | 21.2 | 8.7 | 0.1 | 0.0 |
| | | Generalized Re-Weight | 23.4 | 39.8 | 17.8 | 6.2 | 0.0 | 0.0 |
| | | Balanced Softmax Loss | 30.3 | 41.3 | 26.8 | 18.1 | 16.9 | 2.0 |
| | | Logit Adjustment loss | 31.9 | 40.3 | 29.0 | 23.2 | 20.1 | 2.0 |
| | | LADE loss | 27.3 | 38.8 | 22.7 | 16.5 | 15.2 | 2.0 |
| | PEFT | CE loss | 31.9 | 45.8 | 28.2 | 15.0 | 0.2 | 0.0 |
| | | Focal loss | 30.9 | 45.0 | 26.9 | 14.0 | 0.1 | 0.0 |
| | | LDAM loss | 34.9 | 46.9 | 31.5 | 20.5 | 0.4 | 0.0 |
| | | Class Balanced loss | 32.0 | 45.9 | 28.1 | 15.0 | 0.1 | 0.0 |
| | | Generalized Re-Weight | 34.2 | 47.0 | 32.1 | 15.7 | 0.1 | 0.0 |
| | | Balanced Softmax Loss | 38.3 | 45.3 | 36.6 | 29.5 | 26.7 | 3.0 |
| | | Logit Adjustment loss | 39.7 | 42.2 | 39.7 | 35.2 | 29.9 | 4.0 |
| | | LADE loss | 0.3 | 0.0 | 0.0 | 1.5 | 0.0 | 0.0 |

Table 27: Detailed results of applying different losses to the ImageNet-LT dataset.

| | | | Mean | Many | Med. | Few | Harmonic mean | Worst case |
|---|---|---|---|---|---|---|---|---|
| CLIP-ViT-B/16 | FFT | CE loss | 49.9 | 69.0 | 44.0 | 16.6 | 0.0 | 0.0 |
| | | Focal loss | 48.2 | 67.4 | 41.8 | 16.4 | 0.0 | 0.0 |
| | | LDAM loss | 50.4 | 67.0 | 45.7 | 20.3 | 0.0 | 0.0 |
| | | Class Balanced loss | 50.0 | 68.8 | 43.9 | 18.2 | 0.0 | 0.0 |
| | | Generalized Re-Weight | 49.0 | 67.7 | 42.8 | 18.0 | 0.0 | 0.0 |
| | | Balanced Softmax Loss | 54.6 | 64.8 | 51.0 | 38.2 | 0.3 | 0.0 |
| | | Logit Adjustment loss | 54.0 | 59.8 | 51.5 | 46.5 | 1.0 | 0.0 |
| | | LADE loss | 53.0 | 63.4 | 50.7 | 32.1 | 0.1 | 0.0 |
| | PEFT | CE loss | 70.6 | 85.5 | 67.6 | 38.8 | 0.1 | 0.0 |
| | | Focal loss | 70.1 | 84.8 | 67.1 | 39.1 | 0.3 | 0.0 |
| | | LDAM loss | 71.6 | 85.4 | 69.3 | 40.7 | 0.1 | 0.0 |
| | | Class Balanced loss | 71.2 | 85.5 | 67.7 | 43.2 | 0.5 | 0.0 |
| | | Generalized Re-Weight | 74.5 | 81.8 | 74.2 | 54.6 | 59.8 | 2.0 |
| | | Balanced Softmax Loss | 76.7 | 81.2 | 75.4 | 68.5 | 70.2 | 12.0 |
| | | Logit Adjustment loss | 75.6 | 75.0 | 75.7 | 76.7 | 67.9 | 4.0 |
| | | LADE loss | 76.3 | 81.1 | 75.3 | 66.6 | 69.3 | 8.0 |
| IN21K-ViT-B/16 | FFT | CE loss | 52.1 | 70.1 | 45.9 | 23.0 | 0.1 | 0.0 |
| | | Focal loss | 51.0 | 69.1 | 44.5 | 22.6 | 0.1 | 0.0 |
| | | LDAM loss | 52.2 | 69.6 | 45.8 | 25.2 | 0.1 | 0.0 |
| | | Class Balanced loss | 52.3 | 70.3 | 46.1 | 23.6 | 0.1 | 0.0 |
| | | Generalized Re-Weight | 50.9 | 69.3 | 44.5 | 21.0 | 0.1 | 0.0 |
| | | Balanced Softmax Loss | 55.6 | 68.4 | 51.5 | 35.3 | 36.7 | 0.0 |
| | | Logit Adjustment loss | 56.2 | 66.2 | 52.6 | 40.5 | 1.0 | 0.0 |
| | | LADE loss | 48.4 | 61.3 | 43.1 | 30.7 | 0.3 | 0.0 |
| | PEFT | CE loss | 78.2 | 87.5 | 75.8 | 59.9 | 64.4 | 2.0 |
| | | Focal loss | 77.4 | 86.9 | 74.7 | 59.7 | 63.9 | 2.0 |
| | | LDAM loss | 79.4 | 87.2 | 77.3 | 64.9 | 69.5 | 4.0 |
| | | Class Balanced loss | 78.2 | 87.5 | 75.8 | 60.5 | 64.3 | 2.0 |
| | | Generalized Re-Weight | 78.8 | 87.0 | 77.2 | 61.3 | 66.4 | 4.0 |
| | | Balanced Softmax Loss | 81.2 | 85.6 | 79.8 | 73.6 | 76.3 | 16.0 |
| | | Logit Adjustment loss | 81.6 | 83.7 | 80.6 | 78.7 | 77.6 | 16.0 |
| | | LADE loss | 81.2 | 86.1 | 79.4 | 74.0 | 76.6 | 16.0 |

Table 28: Detailed results of applying different classifiers to the CIFAR100-LT dataset.

| | | | Mean | Many | Med. | Few | Harmonic mean | Worst case |
|---|---|---|---|---|---|---|---|---|
| CLIP-ViT-B/16 | FFT | Linear classifier | 54.6 | 82.9 | 55.9 | 20.1 | 0.0 | 0.0 |
| | | Cosine classifier | 56.4 | 84.3 | 57.1 | 22.9 | 0.0 | 0.0 |
| | | $\tau$-normalized classifier ($\tau = 0.5$) | 55.6 | 83.5 | 56.8 | 21.6 | 0.1 | 0.0 |
| | | $\tau$-normalized classifier ($\tau = 1$) | 55.6 | 83.6 | 56.2 | 22.2 | 0.1 | 0.0 |
| | | $\tau$-normalized classifier ($\tau = 2$) | 54.8 | 83.1 | 55.1 | 21.3 | 0.1 | 0.0 |
| | PEFT | Linear classifier | 71.9 | 90.2 | 75.1 | 46.6 | 56.0 | 7.0 |
| | | Cosine classifier | 72.2 | 90.2 | 74.5 | 48.5 | 37.9 | 1.0 |
| | | $\tau$-normalized classifier ($\tau = 0.5$) | 71.7 | 89.9 | 74.3 | 47.3 | 54.9 | 6.0 |
| | | $\tau$-normalized classifier ($\tau = 1$) | 71.9 | 90.0 | 74.6 | 47.6 | 54.1 | 5.0 |
| | | $\tau$-normalized classifier ($\tau = 2$) | 71.8 | 89.9 | 73.7 | 48.4 | 56.7 | 8.0 |
| IN21K-ViT-B/16 | FFT | Linear classifier | 70.3 | 89.6 | 71.9 | 45.8 | 48.0 | 3.0 |
| | | Cosine classifier | 69.6 | 90.2 | 70.7 | 44.3 | 31.6 | 1.0 |
| | | $\tau$-normalized classifier ($\tau = 0.5$) | 69.3 | 89.6 | 69.0 | 46.1 | 49.0 | 4.0 |
| | | $\tau$-normalized classifier ($\tau = 1$) | 68.9 | 89.9 | 70.1 | 42.9 | 48.3 | 4.0 |
| | | $\tau$-normalized classifier ($\tau = 2$) | 68.8 | 89.5 | 69.7 | 43.6 | 45.8 | 3.0 |
| | PEFT | Linear classifier | 80.7 | 93.5 | 80.9 | 65.4 | 41.2 | 1.0 |
| | | Cosine classifier | 83.9 | 94.8 | 84.1 | 71.0 | 65.0 | 6.0 |
| | | $\tau$-normalized classifier ($\tau = 0.5$) | 80.8 | 93.3 | 80.6 | 66.2 | 41.7 | 1.0 |
| | | $\tau$-normalized classifier ($\tau = 1$) | 80.9 | 93.4 | 80.7 | 66.5 | 42.6 | 1.0 |
| | | $\tau$-normalized classifier ($\tau = 2$) | 81.2 | 93.3 | 81.0 | 67.2 | 58.7 | 3.0 |

Table 29: Detailed results of applying different classifiers to the Places-LT dataset.

| | | | Mean | Many | Med. | Few | Harmonic mean | Worst case |
|---|---|---|---|---|---|---|---|---|
| CLIP-ViT-B/16 | FFT | Linear classifier | 24.9 | 40.7 | 20.1 | 6.8 | 0.0 | 0.0 |
| | | Cosine classifier | 24.9 | 40.9 | 19.9 | 6.6 | 0.0 | 0.0 |
| | | $\tau$-normalized classifier ($\tau = 0.5$) | 24.7 | 40.9 | 19.8 | 6.2 | 0.0 | 0.0 |
| | | $\tau$-normalized classifier ($\tau = 1$) | 24.6 | 41.3 | 19.2 | 6.0 | 0.0 | 0.0 |
| | | $\tau$-normalized classifier ($\tau = 2$) | 23.5 | 40.3 | 17.8 | 5.9 | 0.0 | 0.0 |
| | PEFT | Linear classifier | 39.8 | 53.9 | 35.9 | 22.5 | 0.1 | 0.0 |
| | | Cosine classifier | 40.6 | 55.2 | 35.9 | 24.2 | 0.2 | 0.0 |
| | | $\tau$-normalized classifier ($\tau = 0.5$) | 40.3 | 54.9 | 36.1 | 22.8 | 0.1 | 0.0 |
| | | $\tau$-normalized classifier ($\tau = 1$) | 40.0 | 54.7 | 35.4 | 23.4 | 0.2 | 0.0 |
| | | $\tau$-normalized classifier ($\tau = 2$) | 37.6 | 53.1 | 32.8 | 20.2 | 0.1 | 0.0 |
| IN21K-ViT-B/16 | FFT | Linear classifier | 26.0 | 41.4 | 20.9 | 9.5 | 0.1 | 0.0 |
| | | Cosine classifier | 27.1 | 43.3 | 21.8 | 9.1 | 0.1 | 0.0 |
| | | $\tau$-normalized classifier ($\tau = 0.5$) | 25.8 | 41.5 | 20.6 | 8.7 | 0.1 | 0.0 |
| | | $\tau$-normalized classifier ($\tau = 1$) | 25.4 | 41.3 | 20.1 | 8.1 | 0.0 | 0.0 |
| | | $\tau$-normalized classifier ($\tau = 2$) | 24.8 | 41.8 | 19.2 | 6.3 | 0.0 | 0.0 |
| | PEFT | Linear classifier | 31.9 | 45.8 | 28.2 | 15.0 | 0.2 | 0.0 |
| | | Cosine classifier | 38.1 | 53.4 | 33.7 | 20.2 | 0.4 | 0.0 |
| | | $\tau$-normalized classifier ($\tau = 0.5$) | 32.1 | 46.4 | 28.2 | 14.7 | 0.1 | 0.0 |
| | | $\tau$-normalized classifier ($\tau = 1$) | 32.3 | 47.3 | 27.9 | 14.9 | 0.1 | 0.0 |
| | | $\tau$-normalized classifier ($\tau = 2$) | 32.1 | 47.6 | 27.2 | 14.7 | 0.1 | 0.0 |

Table 30: Detailed results of applying different classifiers to the ImageNet-LT dataset.

| | | | Mean | Many | Med. | Few | Harmonic mean | Worst case |
|---|---|---|---|---|---|---|---|---|
| CLIP-ViT-B/16 | FFT | Linear classifier | 49.9 | 69.0 | 44.0 | 16.6 | 0.0 | 0.0 |
| | | Cosine classifier | 50.0 | 69.6 | 43.7 | 16.8 | 0.1 | 0.0 |
| | | $\tau$-normalized classifier ($\tau = 0.5$) | 49.8 | 69.0 | 43.8 | 16.5 | 0.0 | 0.0 |
| | | $\tau$-normalized classifier ($\tau = 1$) | 49.0 | 68.5 | 42.7 | 16.2 | 0.0 | 0.0 |
| | | $\tau$-normalized classifier ($\tau = 2$) | 45.8 | 66.0 | 38.4 | 14.5 | 0.0 | 0.0 |
| | PEFT | Linear classifier | 70.6 | 85.5 | 67.6 | 38.8 | 0.1 | 0.0 |
| | | Cosine classifier | 70.5 | 85.4 | 67.0 | 40.6 | 0.2 | 0.0 |
| | | $\tau$-normalized classifier ($\tau = 0.5$) | 70.5 | 85.5 | 67.3 | 39.5 | 0.2 | 0.0 |
| | | $\tau$-normalized classifier ($\tau = 1$ | 70.1 | 85.4 | 66.6 | 39.5 | 0.2 | 0.0 |
| | | $\tau$-normalized classifier ($\tau = 2$) | 67.2 | 84.0 | 63.0 | 34.8 | 0.2 | 0.0 |
| IN21K-ViT-B/16 | FFT | Linear classifier | 52.1 | 70.1 | 45.9 | 23 | 0.1 | 0.0 |
| | | Cosine classifier | 54.4 | 72.7 | 48.4 | 23.6 | 0.1 | 0.0 |
| | | $\tau$-normalized classifier ($\tau = 0.5$) | 51.6 | 69.4 | 45.4 | 23 | 0.1 | 0.0 |
| | | $\tau$-normalized classifier ($\tau = 1$) | 50.5 | 69.1 | 44.1 | 20.1 | 0.1 | 0.0 |
| | | $\tau$-normalized classifier ($\tau = 2$) | 49.4 | 68.9 | 42.3 | 18.8 | 0.1 | 0.0 |
| | PEFT | Linear classifier | 78.2 | 87.5 | 75.8 | 59.9 | 64.4 | 2.0 |
| | | Cosine classifier | 80.2 | 88.9 | 78.1 | 63.1 | 0.5 | 0.0 |
| | | $\tau$-normalized classifier ($\tau = 0.5$) | 76.9 | 86.9 | 74.4 | 57.4 | 61.7 | 2.0 |
| | | $\tau$-normalized classifier ($\tau = 1$) | 75.5 | 86.3 | 72.6 | 54.7 | 59.4 | 2.0 |
| | | $\tau$-normalized classifier ($\tau = 2$) | 74.6 | 85.5 | 71.3 | 55.4 | 1.0 | 0.0 |

Table 31: Detailed results of applying knowledge distillation to the CIFAR100-LT dataset.

| | | | | Mean | Many | Med. | Few | Harmonic mean | Worst case |
|---|---|---|---|---|---|---|---|---|---|
| Teacher | IN21K-ViT-B/16 | PEFT | | 88.8 | 91.8 | 88.0 | 86.3 | 81.4 | 9.0 |
| Student | DeiT-S | FFT | Baseline | 67.3 | 88.9 | 67.9 | 41.5 | 29.4 | 1.0 |
| | | | distillation | 70.4 | 91.0 | 71.4 | 45.3 | 31.4 | 1.0 |
| | | PEFT | Baseline | 69.9 | 89.5 | 70.5 | 46.4 | 0.1 | 0.0 |
| | | | distillation | 70.0 | 89.3 | 70.4 | 47.0 | 0.1 | 0.0 |
| | DeiT-Ti | FFT | Baseline | 58.7 | 84.3 | 60.1 | 27.2 | 26.6 | 2.0 |
| | | | distillation | 61.7 | 86.7 | 62.3 | 31.7 | 24.5 | 1.0 |
| | | PEFT | Baseline | 60.8 | 84.3 | 61.6 | 32.6 | 0.1 | 0.0 |
| | | | distillation | 60.6 | 84.3 | 61.3 | 32.3 | 0.0 | 0.0 |

Table 32: Detailed results of applying knowledge distillation to the Places-LT dataset.

| | | | | Mean | Many | Med. | Few | Harmonic mean | Worst case |
|---|---|---|---|---|---|---|---|---|---|
| Teacher | CLIP-ViT-B/16 | PEFT | | 51.5 | 50.9 | 52.2 | 50.9 | 37.1 | 1.0 |
| Student | DeiT-S | FFT | Baseline | 27.1 | 43.0 | 22.5 | 7.9 | 0.1 | 0.0 |
| | | | distillation | 30.2 | 46.5 | 25.6 | 10.8 | 0.1 | 0.0 |
| | | PEFT | Baseline | 32.1 | 48.4 | 27.3 | 13.1 | 0.1 | 0.0 |
| | | | distillation | 32.5 | 49.0 | 27.5 | 13.8 | 0.1 | 0.0 |
| | DeiT-Ti | FFT | Baseline | 24.6 | 41.1 | 19.5 | 5.9 | 0.0 | 0.0 |
| | | | distillation | 28.6 | 45.2 | 23.8 | 9.0 | 0.1 | 0.0 |
| | | PEFT | Baseline | 29.4 | 45.5 | 24.4 | 11.2 | 0.1 | 0.0 |
| | | | distillation | 30.0 | 46.2 | 25.0 | 11.3 | 0.1 | 0.0 |

Table 33: Detailed results of applying knowledge distillation to the ImageNet-LT dataset.

|  |  |  |  | Mean | Many | Med. | Few | Harmonic mean | Worst case |
|---|---|---|---|---|---|---|---|---|---|
| Teacher | IN21K-ViT-B/16 | PEFT |  | 83.6 | 85.8 | 83.0 | 80.0 | 80.1 | 16.0 |
| Student | DeiT-S | FFT | Baseline | 58.3 | 74.1 | 53.7 | 29.9 | 0.1 | 0.0 |
|  |  |  | distillation | 60.5 | 75.8 | 56.2 | 32.6 | 0.2 | 0.0 |
|  |  | PEFT | Baseline | 74.6 | 84.6 | 72.3 | 54.5 | 1.0 | 0.0 |
|  |  |  | distillation | 74.9 | 84.6 | 72.6 | 55.9 | 57.6 | 2.0 |
|  | DeiT-Ti | FFT | Baseline | 50.8 | 68.7 | 45.2 | 20.2 | 0.0 | 0.0 |
|  |  |  | distillation | 52.7 | 70.3 | 47.2 | 22.4 | 0.1 | 0.0 |
|  |  | PEFT | Baseline | 65.6 | 78.8 | 62.3 | 40.2 | 0.5 | 0.0 |
|  |  |  | distillation | 65.9 | 78.9 | 62.6 | 40.5 | 1.0 | 0.0 |

Table 34: Detailed results of applying ensemble learning to the CIFAR100-LT dataset.

|  |  |  | Mean | Many | Med. | Few | Harmonic mean | Worst case |
|---|---|---|---|---|---|---|---|---|
| CLIP-ViT-B/16 | FFT | Baseline | 54.6 | 82.9 | 55.9 | 20.1 | 0.0 | 0.0 |
|  |  | Ensemble | 55.6 | 83.7 | 56.4 | 22.0 | 0.1 | 0.0 |
|  | PEFT | Baseline | 71.9 | 90.2 | 75.1 | 46.6 | 56.0 | 7.0 |
|  |  | Ensemble | 76 | 89.4 | 78.8 | 57.1 | 65.7 | 10.0 |
| IN21K-ViT-B/16 | FFT | Baseline | 70.3 | 89.6 | 71.9 | 45.8 | 48.0 | 3.0 |
|  |  | Ensemble | 68.6 | 90.7 | 70.0 | 41.3 | 46.2 | 4.0 |
|  | PEFT | Baseline | 80.7 | 93.5 | 80.9 | 65.4 | 41.2 | 1.0 |
|  |  | Ensemble | 82.2 | 93.6 | 82.9 | 68.1 | 60.7 | 4.0 |

Table 35: Detailed results of applying ensemble learning to the Places-LT dataset.

|  |  |  | Mean | Many | Med. | Few | Harmonic mean | Worst case |
|---|---|---|---|---|---|---|---|---|
| CLIP-ViT-B/16 | FFT | Baseline | 24.9 | 40.7 | 20.1 | 6.8 | 0.0 | 0.0 |
|  |  | Ensemble | 18.9 | 34.8 | 13.4 | 2.4 | 0.0 | 0.0 |
|  | PEFT | Baseline | 39.8 | 53.9 | 35.9 | 22.5 | 0.1 | 0.0 |
|  |  | Ensemble | 45.0 | 55.5 | 43.5 | 28.9 | 0.4 | 0.0 |
| IN21K-ViT-B/16 | FFT | Baseline | 26.0 | 41.4 | 20.9 | 9.5 | 0.1 | 0.0 |
|  |  | Ensemble | 26.7 | 43.1 | 21.4 | 8.6 | 0.1 | 0.0 |
|  | PEFT | Baseline | 31.9 | 45.8 | 28.2 | 15.0 | 0.2 | 0.0 |
|  |  | Ensemble | 36.4 | 49.2 | 33.6 | 19.2 | 17.3 | 1.0 |

Table 36: Detailed results of applying ensemble learning to the ImageNet-LT dataset.

|  |  |  | Mean | Many | Med. | Few | Harmonic mean | Worst case |
|---|---|---|---|---|---|---|---|---|
| CLIP-ViT-B/16 | FFT | Baseline | 49.9 | 69.0 | 44.0 | 16.6 | 0.0 | 0.0 |
|  |  | Ensemble | 36.7 | 54.7 | 29.8 | 9.7 | 0.0 | 0.0 |
|  | PEFT | Baseline | 70.6 | 85.5 | 67.6 | 38.8 | 0.1 | 0.0 |
|  |  | Ensemble | 73.4 | 84.4 | 71.6 | 48.8 | 0.5 | 0.0 |
| IN21K-ViT-B/16 | FFT | Baseline | 52.1 | 70.1 | 45.9 | 23.0 | 0.1 | 0.0 |
|  |  | Ensemble | 54.2 | 71.9 | 48.6 | 24.1 | 0.1 | 0.0 |
|  | PEFT | Baseline | 78.2 | 87.5 | 75.8 | 59.9 | 64.4 | 2.0 |
|  |  | Ensemble | 80.4 | 87.9 | 78.7 | 65.1 | 70.3 | 4.0 |

Table 37: Detailed results of applying mixup to the CIFAR100-LT dataset.

|  |  |  | Mean | Many | Med. | Few | Harmonic mean | Worst case |
|---|---|---|---|---|---|---|---|---|
| CLIP-ViT-B/16 | FFT | Baseline | 51.0 | 70.3 | 51.0 | 30.3 | 36.4 | 7.0 |
|  |  | Mixup | 68.7 | 81.9 | 69.7 | 51.9 | 61.1 | 19.0 |
|  | PEFT | Baseline | 80.1 | 86.5 | 80.0 | 72.9 | 77.5 | 38.0 |
|  |  | Mixup | 79.7 | 82.5 | 80.5 | 75.1 | 78.1 | 21.0 |
| IN21K-ViT-B/16 | FFT | Baseline | 75.8 | 88.7 | 76.4 | 59.9 | 63.0 | 6.0 |
|  |  | Mixup | 81.6 | 86.7 | 82.5 | 74.5 | 73.6 | 8.0 |
|  | PEFT | Baseline | 85.1 | 92.0 | 84.8 | 77.4 | 79.5 | 18.0 |
|  |  | Mixup | 86.7 | 89.3 | 86.2 | 84.1 | 84.0 | 29.0 |

Table 38: Detailed results of applying mixup to the Places-LT dataset.

| | | | Mean | Many | Med. | Few | Harmonic mean | Worst case |
|---|---|---|---|---|---|---|---|---|
| CLIP-ViT-B/16 | FFT | Baseline | 31.3 | 39.7 | 28.0 | 23.3 | 20.6 | 3.0 |
| | | Mixup | 35.8 | 41.8 | 34.6 | 27.6 | 25.5 | 3.0 |
| | PEFT | Baseline | 48.8 | 49.7 | 49.0 | 46.9 | 39.4 | 4.0 |
| | | Mixup | 49.8 | 49.9 | 50.5 | 48.1 | 37.5 | 2.0 |
| IN21K-ViT-B/16 | FFT | Baseline | 30.3 | 41.3 | 26.8 | 18.1 | 16.9 | 2.0 |
| | | Mixup | 33.3 | 42.0 | 30.9 | 23.0 | 21.7 | 3.0 |
| | PEFT | Baseline | 38.3 | 45.3 | 36.6 | 29.5 | 26.7 | 3.0 |
| | | Mixup | 45.0 | 48.1 | 44.9 | 39.8 | 35.2 | 5.0 |

Table 39: Detailed results of applying mixup to the ImageNet-LT dataset.

| | | | Mean | Many | Med. | Few | Harmonic mean | Worst case |
|---|---|---|---|---|---|---|---|---|
| CLIP-ViT-B/16 | FFT | Baseline | 54.6 | 64.8 | 51.0 | 38.2 | 0.3 | 0.0 |
| | | Mixup | 58.7 | 67.9 | 56.8 | 39.1 | 0.2 | 0.0 |
| | PEFT | Baseline | 76.7 | 81.2 | 75.4 | 68.5 | 70.2 | 12.0 |
| | | Mixup | 75.2 | 78.9 | 74.8 | 66.2 | 67.3 | 8.0 |
| IN21K-ViT-B/16 | FFT | Baseline | 55.6 | 68.4 | 51.5 | 35.3 | 36.7 | 0.0 |
| | | Mixup | 61.5 | 72.0 | 57.6 | 45.6 | 1.0 | 0.0 |
| | PEFT | Baseline | 81.2 | 85.6 | 79.8 | 73.6 | 76.3 | 16.0 |
| | | Mixup | 83.3 | 85.1 | 82.6 | 80.8 | 79.2 | 10.0 |

Table 40: Detailed results of applying label smoothing to the CIFAR100-LT dataset.

| | | | Mean | Many | Med. | Few | Harmonic mean | Worst case |
|---|---|---|---|---|---|---|---|---|
| CLIP-ViT-B/16 | FFT | CE | 54.6 | 82.9 | 55.9 | 20.1 | 0.0 | 0.0 |
| | | CE (w/ LS) | 56.2 | 85.2 | 56.8 | 21.7 | 15.4 | 1.0 |
| | | BS | 58.0 | 75.5 | 58.9 | 36.5 | 42.3 | 6.0 |
| | | BS (w/ LS) | 59.8 | 71.4 | 57.0 | 49.6 | 51.0 | 12.0 |
| | PEFT | CE | 71.9 | 90.2 | 75.1 | 46.6 | 56.0 | 7.0 |
| | | CE (w/ LS) | 71.7 | 89.8 | 75.2 | 46.6 | 36.1 | 1.0 |
| | | BS | 80.1 | 86.5 | 80.0 | 72.9 | 77.5 | 38.0 |
| | | BS (w/ LS) | 80.6 | 84.1 | 80.1 | 77.2 | 78.5 | 44.0 |
| IN21K-ViT-B/16 | FFT | CE | 70.3 | 89.6 | 71.9 | 45.8 | 48.0 | 3.0 |
| | | CE (w/ LS) | 71.3 | 91.1 | 72.7 | 46.6 | 42.0 | 2.0 |
| | | BS | 75.8 | 88.7 | 76.4 | 59.9 | 63.0 | 6.0 |
| | | BS (w/ LS) | 78.2 | 90.1 | 76.8 | 65.8 | 71.9 | 23.0 |
| | PEFT | CE | 80.7 | 93.5 | 80.9 | 65.4 | 41.2 | 1.0 |
| | | CE (w/ LS) | 82.7 | 94.5 | 82.5 | 69.2 | 61.7 | 4.0 |
| | | BS | 85.1 | 92.0 | 84.8 | 77.4 | 79.5 | 18.0 |
| | | BS (w/ LS) | 88.1 | 89.5 | 86.6 | 88.2 | 86.6 | 50.0 |

Table 41: Detailed results of applying label smoothing to the Places-LT dataset.

| | | | Mean | Many | Med. | Few | Harmonic mean | Worst case |
|---|---|---|---|---|---|---|---|---|
| CLIP-ViT-B/16 | FFT | CE | 24.7 | 40.5 | 19.8 | 6.8 | 0.1 | 0.0 |
| | | CE (w/ LS) | 25.0 | 40.5 | 19.7 | 8.3 | 0.1 | 0.0 |
| | | BS | 31.3 | 39.7 | 28.0 | 23.3 | 20.6 | 3.0 |
| | | BS (w/ LS) | 28.6 | 31.9 | 25.5 | 29.7 | 0.4 | 0.0 |
| | PEFT | CE | 39.8 | 53.9 | 35.9 | 22.5 | 0.1 | 0.0 |
| | | CE (w/ LS) | 39.7 | 54.6 | 35.7 | 21.2 | 0.0 | 0.0 |
| | | BS | 48.8 | 49.7 | 49.0 | 46.9 | 39.4 | 4.0 |
| | | BS (w/ LS) | 49.4 | 48.9 | 49.7 | 49.4 | 37.9 | 3.0 |
| IN21K-ViT-B/16 | FFT | CE | 26.0 | 41.4 | 20.9 | 9.5 | 0.1 | 0.0 |
| | | CE (w/ LS) | 26.9 | 43.1 | 21.7 | 8.8 | 0.1 | 0.0 |
| | | BS | 30.3 | 41.3 | 26.8 | 18.1 | 16.9 | 2.0 |
| | | BS (w/ LS) | 32.4 | 38.7 | 29.4 | 27.5 | 0.4 | 0.0 |
| | PEFT | CE | 31.9 | 45.8 | 28.2 | 15.0 | 0.2 | 0.0 |
| | | CE (w/ LS) | 34.1 | 48.3 | 30.0 | 17.3 | 0.1 | 0.0 |
| | | BS | 38.3 | 45.3 | 36.6 | 29.5 | 26.7 | 3.0 |
| | | BS (w/ LS) | 41.8 | 44.6 | 41.2 | 37.8 | 30.6 | 5.0 |

Table 42: Detailed results of applying label smoothing to the ImageNet-LT dataset.

| | | | Mean | Many | Med. | Few | Harmonic mean | Worst case |
|---|---|---|---|---|---|---|---|---|
| CLIP-ViT-B/16 | FFT | CE | 49.9 | 69.0 | 44.0 | 16.6 | 0.0 | 0.0 |
| | | CE (w/ LS) | 51.4 | 69.7 | 45.8 | 19.1 | 0.0 | 0.0 |
| | | BS | 54.6 | 64.8 | 51.0 | 38.2 | 0.3 | 0 |
| | | BS (w/ LS) | 55.5 | 63.2 | 52.2 | 45.3 | 41.0 | 4.0 |
| | PEFT | CE | 70.6 | 85.5 | 67.6 | 38.8 | 0.1 | 0.0 |
| | | CE (w/ LS) | 70.5 | 85.7 | 67.7 | 37.5 | 0.1 | 0.0 |
| | | BS | 76.7 | 81.2 | 75.4 | 68.5 | 70.2 | 12.0 |
| | | BS (w/ LS) | 76.7 | 80.1 | 75.7 | 70.3 | 70.5 | 12.0 |
| IN21K-ViT-B/16 | FFT | CE | 52.1 | 70.1 | 45.9 | 23.0 | 0.1 | 0.0 |
| | | CE (w/ LS) | 54.5 | 73.2 | 48.2 | 24.2 | 0.1 | 0.0 |
| | | BS | 55.6 | 68.4 | 51.5 | 35.3 | 36.7 | 0.0 |
| | | BS (w/ LS) | 59.0 | 68.9 | 54.5 | 43.8 | 43.4 | 2.0 |
| | PEFT | CE | 78.2 | 87.5 | 75.8 | 59.9 | 64.4 | 2.0 |
| | | CE (w/ LS) | 80.4 | 88.3 | 78.3 | 65.4 | 66.4 | 2.0 |
| | | BS | 81.2 | 85.6 | 79.8 | 73.6 | 76.3 | 16.0 |
| | | BS (w/ LS) | 83.0 | 85.0 | 82.1 | 80.3 | 79.1 | 16.0 |

Table 43: Ablation experiment on CIFAR100-LT.

| | | Cosine Classifier | Square-root sampling | Balanced Softmax | Label Smoothing | Auto Augment | Mixup | Mean | Many | Med. | Few | Hmean | Worst |
|---|---|---|---|---|---|---|---|---|---|---|---|---|---|
| CLIP-ViT-B/16 | FFT | | | | | | | 46.9 | 75.6 | 46.3 | 14.1 | 10.7 | 1.0 |
| | | ✓ | | | | | | 41.5 | 70.6 | 39.1 | 10.2 | 0.0 | 0.0 |
| | | ✓ | ✓ | | | | | 60.0 | 82.7 | 64.5 | 28.2 | 20.6 | 1.0 |
| | | ✓ | ✓ | ✓ | | | | 65.7 | 80.3 | 69.1 | 44.7 | 51.0 | 6.0 |
| | | ✓ | ✓ | ✓ | ✓ | | | 67.4 | 80.8 | 70.3 | 48.6 | 55.5 | 8.0 |
| | | ✓ | ✓ | ✓ | ✓ | ✓ | | 29.7 | 38.9 | 31.8 | 16.4 | 0.0 | 0.0 |
| | | ✓ | ✓ | ✓ | ✓ | | ✓ | 49.6 | 61.5 | 52.6 | 32.3 | 25.2 | 1.0 |
| | | ✓ | ✓ | ✓ | ✓ | ✓ | ✓ | 14.4 | 15.3 | 17.2 | 10.0 | 0.0 | 0.0 |
| | PEFT | | | | | | | 71.9 | 90.1 | 75.2 | 47.0 | 56.5 | 8.0 |
| | | ✓ | | | | | | 72.8 | 90.2 | 75.5 | 49.2 | 56.2 | 5.0 |
| | | ✓ | ✓ | | | | | 77.0 | 87.7 | 78.9 | 62.3 | 68.7 | 11.0 |
| | | ✓ | ✓ | ✓ | | | | 80.1 | 84.5 | 81.0 | 74.0 | 77.0 | 28.0 |
| | | ✓ | ✓ | ✓ | ✓ | | | 80.5 | 84.1 | 81.1 | 75.7 | 77.9 | 35.0 |
| | | ✓ | ✓ | ✓ | ✓ | ✓ | | 79.3 | 80.9 | 80.2 | 76.4 | 76.4 | 35.0 |
| | | ✓ | ✓ | ✓ | ✓ | | ✓ | 79.3 | 81.2 | 80.1 | 76.2 | 74.5 | 17.0 |
| | | ✓ | ✓ | ✓ | ✓ | ✓ | ✓ | 77.5 | 79.3 | 78.1 | 74.9 | 73.1 | 28.0 |
| IN21K-ViT-B/16 | FFT | | | | | | | 71.1 | 89.3 | 72.6 | 48.0 | 50.6 | 3.0 |
| | | ✓ | | | | | | 71.4 | 91.0 | 73.0 | 46.8 | 39.0 | 2.0 |
| | | ✓ | ✓ | | | | | 75.1 | 91.4 | 76.6 | 54.3 | 53.3 | 3.0 |
| | | ✓ | ✓ | ✓ | | | | 82.7 | 90.6 | 83.1 | 72.9 | 74.4 | 9.0 |
| | | ✓ | ✓ | ✓ | ✓ | | | 81.5 | 91.7 | 81.3 | 69.8 | 75.6 | 16.0 |
| | | ✓ | ✓ | ✓ | ✓ | ✓ | | 82.2 | 91.5 | 82.9 | 70.3 | 76.8 | 23.0 |
| | | ✓ | ✓ | ✓ | ✓ | | ✓ | 83.9 | 91.2 | 83.7 | 75.7 | 77.8 | 15.0 |
| | | ✓ | ✓ | ✓ | ✓ | ✓ | ✓ | 84.7 | 89.5 | 85.3 | 78.4 | 81.6 | 26.0 |
| | PEFT | | | | | | | 81.6 | 93.3 | 81.9 | 67.6 | 41.9 | 1.0 |
| | | ✓ | | | | | | 84.2 | 94.9 | 84.1 | 71.8 | 62.0 | 4.0 |
| | | ✓ | ✓ | | | | | 87.2 | 94.2 | 87.1 | 79.2 | 79.2 | 12.0 |
| | | ✓ | ✓ | ✓ | | | | 89.1 | 92.6 | 88.5 | 85.8 | 86.5 | 28.0 |
| | | ✓ | ✓ | ✓ | ✓ | | | 89.2 | 91.7 | 88.4 | 87.2 | 87.3 | 40.0 |
| | | ✓ | ✓ | ✓ | ✓ | ✓ | | 88.9 | 90.8 | 88.2 | 87.5 | 87.3 | 43.0 |
| | | ✓ | ✓ | ✓ | ✓ | | ✓ | 88.1 | 89.6 | 88.0 | 86.5 | 84.4 | 19.0 |
| | | ✓ | ✓ | ✓ | ✓ | ✓ | ✓ | 87.6 | 88.9 | 87.1 | 86.6 | 84.6 | 24.0 |

Table 44: Ablation experiment on Places-LT.

| | | Cosine Classifier | Square-root sampling | Balanced Softmax | Label Smoothing | Auto Augment | Mixup | Mean | Many | Med. | Few | Hmean | Worst |
|---|---|---|---|---|---|---|---|---|---|---|---|---|---|
| CLIP -ViT -B/16 | FFT | | | | | | | 23.7 | 39.8 | 18.5 | 6.0 | 0.1 | 0.0 |
| | | ✓ | | | | | | 24.5 | 41.0 | 19.2 | 6.3 | 0.0 | 0.0 |
| | | ✓ | ✓ | | | | | 36.9 | 50.9 | 34.4 | 16.7 | 18.7 | 1.0 |
| | | ✓ | ✓ | ✓ | | | | 42.1 | 17.4 | 42.8 | 31.1 | 31.7 | 4.0 |
| | | ✓ | ✓ | ✓ | ✓ | | | 42.2 | 49.2 | 42.5 | 28.6 | 31.3 | 5.0 |
| | | ✓ | ✓ | ✓ | ✓ | ✓ | | 43.7 | 47.3 | 45.3 | 33.3 | 32.0 | 4.0 |
| | | ✓ | ✓ | ✓ | ✓ | | ✓ | 45.6 | 48.0 | 47.1 | 37.7 | 30.8 | 1.0 |
| | | ✓ | ✓ | ✓ | ✓ | ✓ | ✓ | 45.4 | 46.4 | 47.1 | 39.4 | 32.4 | 5.0 |
| | PEFT | | | | | | | 39.8 | 54.5 | 35.7 | 22.3 | 0.1 | 0.0 |
| | | ✓ | | | | | | 40.6 | 55.1 | 36.2 | 24.0 | 0.1 | 0.0 |
| | | ✓ | ✓ | | | | | 48.0 | 56.1 | 46.0 | 37.7 | 31.8 | 1.0 |
| | | ✓ | ✓ | ✓ | | | | 51.3 | 51.3 | 51.9 | 49.9 | 40.4 | 3.0 |
| | | ✓ | ✓ | ✓ | ✓ | | | 51.2 | 51.2 | 51.9 | 49.8 | 39.0 | 2.0 |
| | | ✓ | ✓ | ✓ | ✓ | ✓ | | 51.1 | 50.6 | 51.9 | 50.1 | 38.8 | 2.0 |
| | | ✓ | ✓ | ✓ | ✓ | | ✓ | 50.7 | 50.6 | 51.3 | 49.6 | 39.1 | 4.0 |
| | | ✓ | ✓ | ✓ | ✓ | ✓ | ✓ | 50.2 | 49.9 | 50.9 | 48.9 | 35.8 | 2.0 |
| IN21K -ViT -B/16 | FFT | | | | | | | 25.7 | 40.9 | 20.5 | 9.4 | 0.1 | 0.0 |
| | | ✓ | | | | | | 26.6 | 42.9 | 21.3 | 8.4 | 0.2 | 0.0 |
| | | ✓ | ✓ | | | | | 38.2 | 51.9 | 34.8 | 21.0 | 19.3 | 2.0 |
| | | ✓ | ✓ | ✓ | | | | 42.3 | 50.3 | 41.1 | 30.0 | 30.1 | 4.0 |
| | | ✓ | ✓ | ✓ | ✓ | | | 42.5 | 51.1 | 41.5 | 29.2 | 29.6 | 3.0 |
| | | ✓ | ✓ | ✓ | ✓ | ✓ | | 43.8 | 50.5 | 43.9 | 31.1 | 31.8 | 3.0 |
| | | ✓ | ✓ | ✓ | ✓ | | ✓ | 45.4 | 50.4 | 46.0 | 34.9 | 33.2 | 3.0 |
| | | ✓ | ✓ | ✓ | ✓ | ✓ | ✓ | 46.0 | 49.9 | 47.0 | 36.6 | 33.4 | 3.0 |
| | PEFT | | | | | | | 31.7 | 45.5 | 27.8 | 15.1 | 0.1 | 0.0 |
| | | ✓ | | | | | | 37.8 | 53.5 | 33.5 | 18.9 | 0.1 | 0.0 |
| | | ✓ | ✓ | | | | | 44.7 | 53.4 | 42.8 | 32.9 | 29.8 | 4.0 |
| | | ✓ | ✓ | ✓ | | | | 48.4 | 49.4 | 49.6 | 43.7 | 36.7 | 4.0 |
| | | ✓ | ✓ | ✓ | ✓ | | | 47.9 | 49.0 | 49.1 | 43.1 | 36.0 | 4.0 |
| | | ✓ | ✓ | ✓ | ✓ | ✓ | | 48.2 | 49.3 | 49.3 | 43.5 | 36.6 | 5.0 |
| | | ✓ | ✓ | ✓ | ✓ | | ✓ | 48.4 | 48.3 | 49.4 | 46.0 | 34.0 | 1.0 |
| | | ✓ | ✓ | ✓ | ✓ | ✓ | ✓ | 48.3 | 47.7 | 49.6 | 46.2 | 34.9 | 2.0 |

Table 45: Ablation experiment on ImageNet-LT.

| | | Cosine Classifier | Square-root sampling | Balanced Softmax | Label Smoothing | Auto Augment | Mixup | Mean | Many | Med. | Few | Hmean | Worst |
|---|---|---|---|---|---|---|---|---|---|---|---|---|---|
| CLIP -ViT -B/16 | FFT | | | | | | | 48.7 | 67.9 | 42.5 | 16.0 | 0.0 | 0.0 |
| | | ✓ | | | | | | 48.7 | 68.6 | 42.2 | 15.5 | 0.0 | 0.0 |
| | | ✓ | ✓ | | | | | 60.1 | 75.0 | 56.0 | 32.1 | 0.1 | 0.0 |
| | | ✓ | ✓ | ✓ | | | | 63.2 | 71.9 | 60.9 | 46.6 | 1.0 | 0.0 |
| | | ✓ | ✓ | ✓ | ✓ | | | 64.1 | 73.1 | 61.6 | 47.4 | 51.2 | 4.0 |
| | | ✓ | ✓ | ✓ | ✓ | ✓ | | 64.5 | 71.4 | 63.0 | 50.1 | 51.9 | 2.0 |
| | | ✓ | ✓ | ✓ | ✓ | | ✓ | 65.7 | 72.0 | 64.2 | 53.2 | 54.6 | 6.0 |
| | | ✓ | ✓ | ✓ | ✓ | ✓ | ✓ | 63.9 | 70.0 | 62.7 | 51.0 | 50.1 | 2.0 |
| | PEFT | | | | | | | 70.5 | 85.5 | 67.5 | 38.3 | 0.1 | 0.0 |
| | | ✓ | | | | | | 70.4 | 85.5 | 67.0 | 39.9 | 0.1 | 0.0 |
| | | ✓ | ✓ | | | | | 74.7 | 84.0 | 72.7 | 55.4 | 60.8 | 4.0 |
| | | ✓ | ✓ | ✓ | | | | 77.0 | 80.8 | 75.9 | 69.6 | 71.1 | 14.0 |
| | | ✓ | ✓ | ✓ | ✓ | | | 77.2 | 80.5 | 76.3 | 71.5 | 71.2 | 14.0 |
| | | ✓ | ✓ | ✓ | ✓ | ✓ | | 76.6 | 79.3 | 75.9 | 71.2 | 69.8 | 6.0 |
| | | ✓ | ✓ | ✓ | ✓ | | ✓ | 75.5 | 78.3 | 74.6 | 70.4 | 68.1 | 8.0 |
| | | ✓ | ✓ | ✓ | ✓ | ✓ | ✓ | 74.9 | 77.3 | 74.4 | 69.8 | 66.6 | 6.0 |
| IN21K -ViT -B/16 | FFT | | | | | | | 50.8 | 69.1 | 44.4 | 21.8 | 0.1 | 0.0 |
| | | ✓ | | | | | | 53.1 | 71.4 | 46.9 | 23.0 | 0.1 | 0.0 |
| | | ✓ | ✓ | | | | | 71.5 | 82.3 | 68.5 | 51.5 | 55.8 | 2.0 |
| | | ✓ | ✓ | ✓ | | | | 73.4 | 81.3 | 71.0 | 59.3 | 65.5 | 6.0 |
| | | ✓ | ✓ | ✓ | ✓ | | | 75.2 | 82.1 | 73.1 | 62.8 | 67.7 | 10.0 |
| | | ✓ | ✓ | ✓ | ✓ | ✓ | | 75.5 | 81.9 | 74.0 | 62.8 | 68.4 | 10.0 |
| | | ✓ | ✓ | ✓ | ✓ | | ✓ | 76.4 | 82.1 | 74.8 | 65.7 | 69.8 | 10.0 |
| | | ✓ | ✓ | ✓ | ✓ | ✓ | ✓ | 77.0 | 82.1 | 75.7 | 67.1 | 70.8 | 10.0 |
| | PEFT | | | | | | | 78.2 | 87.4 | 76.0 | 59.8 | 1.0 | 0.0 |
| | | ✓ | | | | | | 80.3 | 88.8 | 78.2 | 63.6 | 0.5 | 0.0 |
| | | ✓ | ✓ | | | | | 82.6 | 88.1 | 81.3 | 71.5 | 75.1 | 6.0 |
| | | ✓ | ✓ | ✓ | | | | 83.6 | 86.4 | 83.0 | 78.2 | 79.6 | 10.0 |
| | | ✓ | ✓ | ✓ | ✓ | | | 84.1 | 85.8 | 83.6 | 80.6 | 80.2 | 16.0 |
| | | ✓ | ✓ | ✓ | ✓ | ✓ | | 84.1 | 85.8 | 83.6 | 80.9 | 80.2 | 14.0 |
| | | ✓ | ✓ | ✓ | ✓ | | ✓ | 84.1 | 85.1 | 83.8 | 82.8 | 80.1 | 12.0 |
| | | ✓ | ✓ | ✓ | ✓ | ✓ | ✓ | 84.2 | 85.1 | 83.9 | 82.8 | 80.3 | 14.0 |

Table 46: Ablation experiment on iNaturalist 2018.

| | | Cosine Classifier | Square-root sampling | Balanced Softmax | Label Smoothing | Auto Augment | Mixup | Mean | Many | Med. | Few | Hmean | Worst |
|---|---|---|---|---|---|---|---|---|---|---|---|---|---|
| CLIP-ViT-B/16 | FFT | ✓ | | | | | | 58.4 | 73.6 | 62.6 | 49.1 | 0.0 | 0.0 |
| | | ✓ | ✓ | | | | | 63.3 | 72.2 | 64.6 | 59.5 | 0.0 | 0.0 |
| | | ✓ | ✓ | ✓ | | | | 68.4 | 70.3 | 69.4 | 66.7 | 0.0 | 0.0 |
| | | ✓ | ✓ | ✓ | ✓ | | | 70.9 | 66.1 | 71.3 | 71.5 | 0.0 | 0.0 |
| | | ✓ | ✓ | ✓ | ✓ | ✓ | | 71.5 | 65.1 | 71.9 | 72.7 | 0.0 | 0.0 |
| | | ✓ | ✓ | ✓ | ✓ | ✓ | ✓ | 69.6 | 61.3 | 69.9 | 71.4 | 0.0 | 0.0 |
| | | ✓ | ✓ | ✓ | ✓ | | ✓ | 69.4 | 59.9 | 69.6 | 71.6 | 0.0 | 0.0 |
| | | ✓ | ✓ | ✓ | ✓ | ✓ | ✓ | 48.7 | 34.2 | 47.2 | 54.4 | 0.0 | 0.0 |
| | PEFT | ✓ | | | | | | 69.5 | 82.1 | 73.1 | 61.7 | 0.0 | 0.0 |
| | | ✓ | ✓ | | | | | 75.3 | 81.6 | 76.0 | 72.7 | 0.0 | 0.0 |
| | | ✓ | ✓ | ✓ | | | | 76.8 | 78.6 | 77.4 | 75.5 | 0.0 | 0.0 |
| | | ✓ | ✓ | ✓ | ✓ | | | 79.3 | 73.5 | 79.1 | 81.0 | 0.0 | 0.0 |
| | | ✓ | ✓ | ✓ | ✓ | ✓ | | 79.0 | 73.0 | 78.9 | 80.6 | 0.0 | 0.0 |
| | | ✓ | ✓ | ✓ | ✓ | ✓ | ✓ | 78.3 | 72.0 | 78.4 | 79.8 | 0.0 | 0.0 |
| | | ✓ | ✓ | ✓ | ✓ | | ✓ | 76.9 | 69.1 | 77.0 | 78.8 | 0.0 | 0.0 |
| | | ✓ | ✓ | ✓ | ✓ | ✓ | ✓ | 74.6 | 66.3 | 74.4 | 76.9 | 0.0 | 0.0 |
| IN21K-ViT-B/16 | FFT | ✓ | | | | | | 57.8 | 65.3 | 59.1 | 54.2 | 0.0 | 0.0 |
| | | ✓ | ✓ | | | | | 61.5 | 70.3 | 62.9 | 57.5 | 0.0 | 0.0 |
| | | ✓ | ✓ | ✓ | | | | 72.3 | 75.0 | 73.4 | 70.1 | 0.0 | 0.0 |
| | | ✓ | ✓ | ✓ | ✓ | | | 75.0 | 70.0 | 75.7 | 75.4 | 0.0 | 0.0 |
| | | ✓ | ✓ | ✓ | ✓ | ✓ | | 74.6 | 69.8 | 75.1 | 75.2 | 0.0 | 0.0 |
| | | ✓ | ✓ | ✓ | ✓ | ✓ | ✓ | 74.9 | 68.8 | 75.5 | 75.7 | 0.0 | 0.0 |
| | | ✓ | ✓ | ✓ | ✓ | | ✓ | 73.3 | 65.7 | 73.9 | 74.6 | 0.0 | 0.0 |
| | | ✓ | ✓ | ✓ | ✓ | ✓ | ✓ | 72.3 | 63.5 | 72.9 | 73.9 | 0.0 | 0.0 |
| | PEFT | ✓ | | | | | | 73.6 | 79.2 | 75.8 | 69.5 | 0.0 | 0.0 |
| | | ✓ | ✓ | | | | | 75.6 | 81.2 | 77.2 | 72.2 | 0.0 | 0.0 |
| | | ✓ | ✓ | ✓ | | | | 79.0 | 80.6 | 80.2 | 77.1 | 0.0 | 0.0 |
| | | ✓ | ✓ | ✓ | ✓ | | | 81.1 | 75.6 | 81.7 | 81.9 | 0.1 | 0.0 |
| | | ✓ | ✓ | ✓ | ✓ | ✓ | | 81.1 | 75.8 | 81.8 | 81.7 | 0.1 | 0.0 |
| | | ✓ | ✓ | ✓ | ✓ | ✓ | ✓ | 81.1 | 74.6 | 81.8 | 81.8 | 0.1 | 0.0 |
| | | ✓ | ✓ | ✓ | ✓ | | ✓ | 79.9 | 71.9 | 80.4 | 81.4 | 0.0 | 0.0 |
| | | ✓ | ✓ | ✓ | ✓ | ✓ | ✓ | 79.4 | 71.2 | 79.9 | 80.8 | 0.0 | 0.0 |

Table 47: Comparison of LIFT and our method on accuracy and training cost. "S/E" represents the number of training samples in each epoch, and "Samples" represents the total number of training samples.

| Datasets | | Acc | Epochs | S/E | Samples |
|---|---|---|---|---|---|
| CIFAR100-IR100 | LIFT | 80.3 | 10 | 10.8K | 108K |
| | Ours | **80.5** | 50 | 1.9K | **95K(↓)** |
| Places-LT | LIFT | **51.5** | 10 | 62.5K | 625K |
| | Ours | 51.2 | 50 | 8.2K | **410K(↓)** |
| ImageNet-LT | LIFT | 77.0 | 10 | 117.0K | 1.17M |
| | Ours | **77.2** | 50 | 20.7K | **1.03M(↓)** |
| iNaturalist 2018 | LIFT | **79.1** | 20 | 437.5K | 8.75M |
| | Ours | 79.0 | 100 | 65.0K | **6.5M(↓)** |

Table 48: Accuracy of our method with DINO.

| Datasets | Overall | Many | Med. | Few |
|---|---|---|---|---|
| CIFAR100-IR100 | 80.3 | 85.1 | 81.7 | 73.0 |
| Places-LT | 43.9 | 45.5 | 45.4 | 37.3 |
| ImageNet-LT | 73.5 | 77.6 | 72.8 | 64.0 |

