# OpenReview forum: "How Does Fine-Tuned Foundation Models Help for Long-Tailed Data"
_ICLR.cc/2026/Conference — Submitted to ICLR 2026_

### Official Review · Reviewer_WP1F · 2025-10-31

**Soundness:** 2
**Presentation:** 3
**Contribution:** 2
**Rating:** 4
**Confidence:** 4

**Summary:**

The paper systematically studies how classic long-tailed learning methods perform when applied to fine-tuning foundation vision models. It evaluates seven categories of techniques (re-sampling, augmentation, class-balanced losses, classifier design, etc.) under both full and parameter-efficient fine-tuning. The authors find that many traditional methods do not transfer well, while a combination of cosine classifier, square-root sampling, Balanced Softmax/logit adjustment, and label smoothing works reliably. They conclude with a unified fine-tuning framework that outperforms prior long-tail methods across multiple benchmarks.

**Strengths:**

1 The paper provides a comprehensive and systematic empirical study of seven major categories of long-tailed learning methods on foundation models

2 Based on extensive experiments, the authors deliver a well-validated unified fine-tuning framework that consistently improves long-tailed performance across multiple datasets and backbones

**Weaknesses:**

(1) I do not fully agree with the authors’ claim in the introduction that “to the best of our knowledge, there has not been a systematic study on how to fine-tune foundation models under a long-tailed distribution.” In fact, LIFT [a] has already provided a systematic analysis of imbalance issues under long-tailed settings and explored various strategies. Works such as [b] and [c] have also examined biases in foundation models under long-tailed distributions, and LPT [d] offers a deeper investigation as well. I recommend that the authors reconsider the positioning of their contribution and more precisely articulate the gap their work aims to fill, rather than relying on an overly broad claim.

(2) The results on several benchmarks do not appear to surpass LIFT, which achieves competitive performance with only 10 training epochs and minimal additional techniques. From this standpoint, it is difficult to assess the practical significance and novelty of the proposed method.

(3) Many of the examined techniques and strategies depend heavily on training hyperparameters such as the number of epochs, learning rate. More experiments are required to understand how sensitive the proposed tricks are to these hyperparameters and to more fully validate their robustness.

[a]  Shi, Jiang-Xin, et al. "Long-tail learning with foundation model: Heavy fine-tuning hurts." arXiv preprint arXiv:2309.10019 (2023). ICML2024

[b] Chen, Jiahao, et al. "Rethinking the Bias of Foundation Model under Long-tailed Distribution." arXiv preprint arXiv:2501.15955 (2025). ICML2025

[c] Wen, Xin, et al. "What makes clip more robust to long-tailed pre-training data? a controlled study for transferable insights." Advances in Neural Information Processing Systems 37 (2024): 36567-36601.

[d] Dong, Bowen, et al. "Lpt: Long-tailed prompt tuning for image classification." arXiv preprint arXiv:2210.01033 (2022). ICLR

**Questions:**

see weakness

---

> ### Author Response · Authors · 2025-11-19
> **Response to Reviewer WP1F**
>
> We are grateful for your positive assessment and insightful comments. We now conduct additional comparative experiments that provide compelling evidence to address your concerns, and we genuinely hope that these responses will help you reconsider the contributions of our paper.
>
> **Question #1: About our claim.**
>
> **Response #1:** Thank you for your suggestion. Indeed, some of the work has already studied fine-tuning strategies for foundation models under long-tailed distributions. However, there is no research on the transfer performance of traditional long-tailed methods based on foundation models. Our work aims to fill the existing gaps in this area. We will revise our writing in the next version to avoid overstating our findings.
>
> **Question #2: Comparison with LIFT.**
>
> **Response #2:** The performance of our model is comparable to that achieved by LIFT, as shown in the following tables. Although we have more epochs, due to the sampling strategy of the data, the total training cost is significantly lower compared to LIFT, **with an average saved cost of 21%** (specific values vary depending on the dataset). Notably, it achieves a remarkable 34% reduction on the Places-LT, demonstrating the effectiveness of our method.
>
> **Table**: Comparison of LIFT and our method on accuracy and training cost. "S/E" represents the number of training samples in each epoch, "Samples" represents the total number of training samples.
>
> | CIFAR100-IR100 | Acc | Epochs | S/E | Samples |
> | :----- | :------: | :-----: | :------: | :-----: |
> | LIFT | 80.3 | 10 | 10.8K | 108K |
> | Ours | **80.5** | 50 | 1.9K | **95K** (↓) |
>
> | Places-LT | Acc | Epochs | S/E | Samples |
> | :------ | :------: | :-----: | :------: | :-----: |
> | LIFT | **51.5** | 10 | 62.5K | 625K |
> | Ours | 51.2 | 50 | 8.2K | **410K** (↓) |
>
> | ImageNet-LT | Acc | Epochs | S/E | Samples |
> | :------ | :------: | :-----: | :------: | :-----: |
> | LIFT | 77.0 | 10 | 117.0K | 1.17M |
> | Ours | **77.2** | 50 | 20.7K | **1.03M** (↓) |
>
> | iNaturalist | Acc | Epochs | S/E | Samples |
> | :------ | :------: | :-----: | :------: | :-----: |
> | LIFT | **79.1** | 20 | 437.5K | 8.75M |
> | Ours | 79.0 | 100 | 65.0K | **6.5M** (↓) |
>
> **Question #3: How sensitive is the proposed framework to hyperparameters?**
>
> **Response #3:** The number of epochs and learning rate are carefully selected. We conduct comprehensive ablation studies on the epochs and learning rates across the CIFAR100-IR100, Places-LT, and ImageNet-LT datasets. The results demonstrate the robustness of our framework to hyperparameter selection. Generally, setting the number of epochs between 30 and 90, and the learning rate between 0.005 and 0.01 can achieve satisfactory results.
>
>
> **Table**: Comparison of the number of epochs and learning rate.
>
> | Epochs | CIFAR100-IR100 | Places-LT | ImageNet-LT
> | :-----: | :------: | :-----: | :------: |
> | 10 | 78.7 | 50.7 | 75.7 |
> | 30 | 80.2 | **51.3** | 77.0 |
> | 50 | **80.8** | 51.2 | **77.2** |
> | 70 | 80.7 | 50.8 | 77.0 |
> | 90 | 80.4 | 50.4 | 76.8 |
>
> | LR | CIFAR100-IR100 | Places-LT | ImageNet-LT
> | :-----: | :------: | :-----: | :------: |
> | 0.001 | 78.7 | 49.8 | 74.4 |
> | 0.005 | **80.8** | **51.5** | 76.9 |
> | 0.01 | **80.8** | 51.2 | **77.2** |
> | 0.05 | 77.9 | 48.9 | 75.7 |
> | 0.1 | 77.1 | 48.3 | 75.3 |
>
> We will update the manuscript based on the above responses in the next few days.

---

> > ### Comment · Reviewer_WP1F · 2025-11-20
> >
> > thanks for your response. i still have two concerns
> >
> > (1) The performance of the proposed method is similar to LIFT, while LIFT is simple enough. It seems that the exploration "transfer performance of traditional long-tailed methods based on foundation models" is meaningless. In addition, what about the average training times between the proposed methods and LIFT? I think this metric is more important.
> >
> > (2) The proposed method is quite sensitive to the number of training epochs and the learning rate, while traditional long-tail methods also have some hyperparameters. Therefore, it seems difficult to use this method in practical applications due to the large number of hyperparameters.

---

> > > ### Author Response · Authors · 2025-11-21
> > > **Additional Responses**
> > >
> > > **Response to concern (1):** We acknowledge that LIFT is simple and straightforward, but the work of LIFT still leaves many areas unexplored, such as sampling strategies, data augmentation, knowledge distillation, ensemble learning, and label smoothing. These methods have shown satisfactory results in traditional long-tail learning, but their effectiveness in foundation models has not been explored yet.
> > >
> > > Our method is meaningful in bridging this gap, as it significantly reduces the number of training samples, given that the training time is proportional to the number of training samples.
> > > For example, under the PEFT setting of the CLIP-ViT/B-16 model, our method requires only 23 minutes to train on the Places-LT dataset, representing a 30% increase in efficiency over LIFT, which takes approximately 30 minutes.
> > >
> > >
> > >
> > >
> > > **Response to concern (2):** Firstly, the statement "quite sensitive to the number of training epochs and the learning rate" is unreasonable.
> > > For example, Figure 4(a) in the paper of LIFT[1] demonstrates the effect of different learning rates for accuracy, and Table 14 in the appendix of LIFT[1] shows the impact of varying epochs on iNaturalist 2018.
> > > The fluctuation magnitude in accuracy observed in these experiments **is identical to that of our method**, so this observation can not support the statement "quite sensitive to the number of training epochs and the learning rate".
> > >
> > > Secondly, regarding the hyperparameters of the long-tail methods, although we use many methods with hyperparameters for ablation experiments, we carefully considered them when building the framework. Our final framework adopts the **hyperparameter-free** method of square-root and BS loss, as well as the same classifier CosClassifier in LIFT. Additionally, LIFT introduces additional hyperparameters by applying the LA loss as the loss function. **Therefore, our method ultimately has no more hyperparameters than LIFT**.
> > >
> > > In addition, our method maintains identical hyperparameters across all datasets and requires no extra tuning, just like LIFT (except for the number of epochs, an adjustment also made by LIFT).
> > >
> > > [1] Shi, Jiang-Xin, et al. "Long-tail learning with foundation model: Heavy fine-tuning hurts." arXiv preprint arXiv:2309.10019 (2023). ICML2024

---

> > > > ### Comment · Reviewer_WP1F · 2025-11-21
> > > >
> > > > Thanks for your response. I agree that "These methods have shown satisfactory results in traditional long-tail learning, but their effectiveness in foundation models has not been explored yet." I also think this exploration is important and very happy to see these explorations under long-tailed tasks. However, i still cannot approve that your performance is very similar to LIFT, only improving the efficiency.
> > > >
> > > > From my perspective, the author's exploration is meaningful, but it's clear that it hasn't yielded satisfactory results yet, and further exploration is necessary.

---

### Official Review · Reviewer_QeSQ · 2025-10-31

**Soundness:** 3
**Presentation:** 2
**Contribution:** 2
**Rating:** 4
**Confidence:** 3

**Summary:**

This paper systematically evaluates long-tail learning methods, including re-sampling and loss functions for fine-tuning foundation models, i.e., CLIP and ViT. A unified framework that outperforms existing approaches on imbalanced datasets has been proposed, and empirical guidelines have been provided for the long-tailed learning community. However, evaluations have been carried out on limited model architectures, and the proposition is mainly validated through accuracy and efficiency. The impact on the learned representations remains undiscovered.

**Strengths:**

- The paper provides a systematic empirical study of long-tail learning methods and offers actionable insights, e.g., recommending Balanced Softmax and Square-root sampling for fine-tuning foundation models on imbalanced data, which is valuable for the whole long-tailed learning community.

-  Proposes a novel framework combining optimal methods to achieve trade-offs between performance and computational cost.

- Tests on 4 datasets with detailed observations, e.g., hyperparameter sensitivity analysis, examine robustness of methods like LADE, noting instability with improper hyperparameters.

**Weaknesses:**

- Only CLIP/ViT are considered. Extending to other architectures (e.g., DINO) could strengthen the generalizability.

- Mentions potential leakage between ImageNet and IN21K-ViT, but doesn’t quantify how its impact was mitigated. Additionally, the BALLAD baseline is omitted in Table 10, which makes superiority claims less convincing.

- The motivation is intuitive, and the work's unified framework combines existing methods, but groundbreaking algorithmic novelty lacks emphasis.

- Some tables (e.g., resampling results across datasets) could be consolidated for brevity. Moreover, the style of the tables is not unified. Some are full-bordered, and some are three-line tables.

**Questions:**

1. What is the impact brought by the new design to the learned representations? Qualitative results and more detailed ablations are preferred to indicate the effectiveness of the proposed method on learned representations.

2. Why were only CLIP and ViT tested? How about DINO?

3. LADE collapses in Fig. 2. Does this reveal fundamental limitations of logit adjustment for foundation models, or is it fixable via hyperparameter tuning?

4. Beyond acknowledging potential leakage between ImageNet and IN21K-ViT, what specific steps were taken to ensure contamination didn't inflate results?

---

> ### Author Response · Authors · 2025-11-19
> **Response to Reviewer QeSQ**
>
> Thank you for the thoughtful and constructive feedback. We conducted additional experiments and provided detailed responses to address your concerns. We sincerely hope you reconsider the evaluation of our paper after reviewing our response.
>
> **Question #1: What is the impact brought by the new design to the learned representations?**
>
> **Response #1:** We conduct some deeper analyses, including t-SNE visualization and classifier weight norms. Please refer to **the response to Reviewer yvXu**.
>
> **Question #2: Results with DINO.**
>
> **Response #2:** To verify the transferability of our framework, we extended it to DINO and conducted corresponding experiments. The results are shown in the following table.
>
> **Table**: Accuracy of our method with DINO.
> | Datasets | Overall | Many | Med. | Few |
> | :-----: | :------: | :-----: | :------: | :------: |
> | CIFAR100-IR100 | 80.3 | 85.1 | 81.7 | 73.0 |
> | Places-LT | 43.9 | 45.5 | 45.4 |  37.3 |
> | ImageNet-LT | 73.5 | 77.6 | 72.8 | 64.0 |
>
> **Question #3: Does data leakage affect the results?**
>
> **Response #3:** We have taken into account the potential data leakage issue. The categories of data for the pre-training model IN21K-ViT can be referred to at https://storage.googleapis.com/bit_models/imagenet21k_wordnet_lemmas.txt. It can be seen that the vast majority of sample classes in the ImageNet dataset are included.
>
> Therefore, we only included its experimental results in Appendix Table 43, which shows that ImageNet-LT exhibits abnormally high performance on IN21K-ViT. We reported the experimental results for only CLIP-ViT in Table 10 of the main text, which were pre-trained unsupervisedly, so the results are not exaggerated.
>
> **Question #4: Why does LADE collapse?**
>
> **Response #4:** The LADE loss is based on Logit adjustment, which is proven effective for long-tailed distributions and foundation models. This is also validated in our experiments.
> However, the LADE loss function introduces numerous additional assumptions based on logit adjustment, making it overly complex. Therefore, it may only be suitable for specific models, such as CNN models rather than ViT models.
>
> **Question #5: About the BALLAD baseline.**
>
> **Response #5:** BALLAD baseline is **not omitted** in Table 10. Table 10 includes most of the mainstream baselines, including BALLAD, LPT, VL-LTR, etc.  Note that "-" in Table 10 means the related paper hasn't reported the corresponding result, rather than that we didn't list them.
>
> We will update the manuscript based on the above responses in the next few days.

---

### Official Review · Reviewer_b51h · 2025-11-01

**Soundness:** 3
**Presentation:** 3
**Contribution:** 3
**Rating:** 6
**Confidence:** 3

**Summary:**

This paper presents a systematic study of how classical long-tail learning methods perform when applied to foundation models (CLIP and ViT) instead of training from scratch. The research categorizes existing methods into 7 groups: (1) Re-sampling, (2) Data Augmentation, (3) Class-sensitive Loss, (4) Balanced Classifier, (5) Knowledge Distillation, (6) Ensemble Learning, and (7) Other tricks. Through extensive experiments across two fine-tuning paradigms (Full Fine-Tuning and Parameter-Efficient Fine-Tuning) and four standard datasets (CIFAR100-LT, Places-LT, ImageNet-LT, iNaturalist 2018), they provide empirical guidelines for practitioners. The authors then propose a unified framework combining the most effective methods and demonstrate competitive performance compared to state-of-the-art approaches.

**Strengths:**

- The paper provides a timely revisit of how existing methods perform when pre-trained models are adopted, which is practically beneficial.
- The experimental setup is well-structured and thorough seven method categories covering major long-tail learning approaches over four datasets. The paper also provides detailed hyperparameter specifications and comprehensive ablation studies, facilitating reproducibility and follow-up research.
- The work provides actionable insights, clearly showing which methods work best in different settings.
- This work also considers training costs, computational efficiency, and hyperparameter sensitivity. This practical consideration is crucial for real-world deployment.

**Weaknesses:**

- The work is purely empirical. While systematic evaluation has value, the contribution is relatively modest. The proposed ultimate framework is also simply a combination of best-performing existing methods without deeper insight into why these combinations work synergistically.
- No statistical significance testing is reported. It would be beneficial to rule out experimental noise with multiple independent runs and increase the reliability of results.

**Questions:**

- Do the authors expect these findings to generalize to other foundation models beyond CLIP and ViT (e.g., DINOv2, MAE, SigLIP2)?
- What properties of pre-trained representations make them more suitable to certain long-tail learning techniques or hyper-parameters?

---

> ### Author Response · Authors · 2025-11-19
> **Response to Reviewer b51h**
>
> We sincerely appreciate your thoughtful evaluation and particularly valuable suggestions for strengthening our work. We provide our point-by-point responses below.
>
> **Question #1: Provide a deeper insight into these combinations.**
>
> **Response #1:** We conduct some deeper analyses, including t-SNE visualization and classifier weight norms. Please refer to **the response to Reviewer yvXu**.
>
> **Question #2: About the reliability of results.**
>
> **Response #2:** Our experimental results are ensured to be reliable through the use of fixed random seeds, which is also the standard practice adopted by most existing long-tailed learning methods.
>
>
> **Question #3: Can it be extended to other foundation models?**
>
> **Response #3:** To verify the transferability of our framework, we extend it to DINO and conduct corresponding experiments. The results are shown in the following table. we are temporarily unable to report results for DINOv2 due to GPU memory limitations. Our framework can also be readily adapted to MAE and SigLIP, which are planned for a future version.
>
> **Table**: Accuracy of our method with DINO.
> | Datasets | Overall | Many | Med. | Few |
> | :-----: | :------: | :-----: | :------: | :------: |
> | CIFAR100-IR100 | 80.3 | 85.1 | 81.7 | 73.0 |
> | Places-LT | 43.9 | 45.5 | 45.4 |  37.3 |
> | ImageNet-LT | 73.5 | 77.6 | 72.8 | 64.0 |
>
> **Question #4: About properties of pre-trained representations**
>
> **Response #4:** Our motivation is to provide guidance for the use of related methods in the long tail domain, given the widespread use of pre-trained models in various fields.
> The question you raised is very interesting. We believe that pre-trained models have the ability to achieve initial balance in downstream tasks[1]. Therefore, the relevant methods may undermine their original ability. For example, the optimal framework does not include augmentation and mixup because they generate excessively unrealistic samples.
>
>
> We will update the manuscript based on the above reponses in the next few days.
>
> [1] Wen, Xin, et al. "What makes clip more robust to long-tailed pre-training data? a controlled study for transferable insights." Advances in Neural Information Processing Systems 37 (2024): 36567-36601.

---

### Official Review · Reviewer_yvXu · 2025-11-05

**Soundness:** 2
**Presentation:** 3
**Contribution:** 2
**Rating:** 4
**Confidence:** 5

**Summary:**

The paper investigates how classic long-tailed (LT) learning techniques fare when fine-tuning foundation models (CLIP, ViT) under both full fine-tuning (FFT) and parameter-efficient fine-tuning (PEFT). It benchmarks seven families of methods (re-sampling, data augmentation, class-sensitive losses, balanced classifiers, and “other tricks”), and synthesizes an “ultimate framework” combining Cosine classifier + Square-root sampling + Balanced Softmax (BS) + Label Smoothing (LS). This framework yields consistent gains across ImageNet-LT, Places-LT, CIFAR100-LT, and iNaturalist-2018, sometimes surpassing recent LT methods, while noting that naive data augmentation have inconsistent effects on
performance across different datasets and models.

**Strengths:**

1: The paper systematically reviews and tests representative techniques (e.g., ROS/RUS/Square-root, Rand/AutoAugment, Focal/LDAM/CB/BS/LA/LADE, Cosine/τ-norm, mixup/LS) across FFT/PEFT and backbones

**Weaknesses:**

1: While this study presents ample empirical results, the findings remain largely superficial and fail to uncover the intrinsic mechanisms by which long-tailed data distributions influence fine-tuning. Prior works, such as [1] and [2], provide theoretical insights into the geometric properties of contrastive representations learned from balanced versus imbalanced datasets. Other studies [3, 4] investigate the underlying mechanisms of long-tailed learning from an empirical perspective. In the context of CLIP, [5] offers a promising direction for exploring how fine-tuning with long-tailed data impacts downstream performance. Incorporating such theoretical or representational analyses would substantially deepen the paper’s contribution and explanatory power.

[1] Dissecting supervised contrastive learning, Graf et al., ICML 2021.

[2] Geometry of Long-Tailed Representation Learning: Rebalancing Features for Skewed Distributions, Yi et al., ICLR 2025.

[3] Imbalance trouble: Revisiting neural-collapse geometry, Thrampoulidis et al., NeurIPS 2022.

[4] What Makes CLIP More Robust to Long-Tailed Pre-Training Data? A Controlled Study for Transferable Insights, Wen et al., NeurIPS 2024.

[5] Decipher the Modality Gap in Multimodal Contrastive Learning: From Convergent Representations to Pairwise Alignment, Yi et al., arxiv.

**Questions:**

1: Why does RUS×N negatively affect tail performance? Beyond reporting the empirical trend, the paper should try to provide a deeper analysis of the underlying mechanism. For instance, can the authors examine representation drift or head-biased margin dynamics as N increases? Such analyses would clarify whether the degradation arises from overfitting to majority classes, loss of feature diversity in tails, or instability in the learned decision boundaries.

2: Please provide a deeper mechanistic analysis of why the combination of Cosine normalization, BS, and LS exhibits robustness under long-tailed (LT) fine-tuning for foundation models (FMs). For instance, an examination of weight norms, feature angular distributions, and class-wise margins before and after fine-tuning would help explain the underlying dynamics contributing to this robustness.

---

> ### Author Response · Authors · 2025-11-19
> **Response to Reviewer yvXu**
>
> We sincerely appreciate your thorough review and valuable feedback on our work and respectfully look forward to your reconsideration after reviewing our responses.
>
> **Question #1: Why does RUS×N negatively affect tail performance?**
>
> **Response #1:** We further compare the models with two different strategies: RUS and RUSx10, using the CLIP-ViT/B-16 architecture under PEFT settings. We extract the features of **tail-class** test data from CIFAR100-IR100 using these two models and visualize the results using t-SNE, as shown below:
>
> **Figure**: t-SNE visualization for RUS and RUSx10. The mean and covariance matrix are computed from the data points, followed by the calculation of the eigenvalues and eigenvectors of the covariance matrix.  The center of the ellipse corresponds to the mean of the data points, the eigenvalues determine the lengths of the major and minor axes, and the eigenvectors define the orientation of the ellipse.
>
> (URL link) t-SNE for RUS：https://ibb.co/bjvP33hL
>
> (URL link) t-SNE for RUSx10: https://ibb.co/9mLS6CC7
>
>
> It can be observed that the decision boundaries (ellipses plot in the t-SNE) for RUS exhibit less overlap. This indicates more distinct decision boundaries for the tail classes, which can indicate better classification performance on tail classes.
>
> Furthermore, we compute the weight norms of the model's classifier and present the results in the following figure. The figure shows that RUS clearly demonstrates a more balanced distribution across all classes.
>
> **Figure**: Classifier Weight norms for RUS and RUSx10.
>
> (URL link) weight norms for RUS: https://ibb.co/7tgYGDz3
>
> **Question #2:  Provide a deeper mechanistic analysis.**
>
> **Response #2:** We follow your suggestion and examine the classifier weight norms, which led to some interesting findings. Specifically, we extract the classifier weight norms from models trained under the PEFT setting of CLIP-ViT/B-16 using four different datasets. The resulting figures show these weight norms against classes arranged along the horizontal axis in descending order of training sample count.
>
>
> **Figure**: Classifier weight norms for Cifar100-IR100 and Places-LT
>
> (URL link) weight norms for CIFAR100-IR100: https://ibb.co/Pz1yccTz
>
> (URL link) weight norms for Places-LT: https://ibb.co/kgMgtxMq
>
> Since the curve representing our method (blue line) overlaps significantly with those of other approaches in the figure, visual assessment alone cannot conclusively determine whether it achieves the best balance. To enable a quantitative comparison, we therefore introduce the standard deviation of the classifier weight norms as a metric to evaluate balance. The results presented in the table below demonstrate that our proposed method indeed achieves the most balanced weight norms.
>
> **Table**: Standard Deviation of classifier weight norms from models trained on different datasets. Each value in the table represents the actual standard deviation when multiplied by $10^{-2}$.
>
> | Standard Deviation ($\times$  $10^{-2}$) | CIFAR100-IR100 | Places-LT | ImageNet-LT | iNaturalist 2018 |
> | :----- | :------: | :-----: | :------: | :------: |
> | CE | 16.3 | 23.8 | 13.6 | 10.6 |
> | CE+Cos | 7.9 | 15.2 | 7.8 | 5.7 |
> | CE+Cos+Sqrt | 1.8 | 3.0 | 1.3 | 3.9  |
> | BS+Cos+Sqrt | 1.8 | 3.0 | 1.3 | 3.9 |
> | BS+Cos+Sqrt+LS | **1.5** | **2.6** | **1.1** | **3.6** |
>
> We will update the manuscript based on the above responses in the next few days.

---

### Author Response · Authors · 2025-11-22
**We have updated the manuscript PDF**

Dear reviewers,

We have made necessary revisions and updated the manuscript PDF. Thank you for your valuable comments and constructive suggestions.

Best,
Authors

---

### Meta-Review · Area_Chair_bEhx · 2026-01-04

**Summary:**

While the reviewers acknowledged the value of the systematic evaluation, they raised significant concerns regarding the depth of analysis, novelty relative to prior work, and generalizability.

Reviewer yvXu and Reviewer QeSQ initially criticized the findings as superficial, requesting deeper mechanistic analyses such as examining representation drift and classifier weight norms to explain why specific strategies like repeated under-sampling degrade performance or why the proposed framework is robust.

 A major point of contention was the paper's contribution. Reviewer WP1F argued that the proposed method offers only incremental improvements over existing baselines like LIFT and remained unconvinced about its practical significance despite the authors' efficiency claims , a sentiment shared by Reviewer b51h who noted the modest nature of the empirical contribution.

 Additionally, Reviewers b51h and QeSQ expressed concerns about the robustness and scope of the experiments, specifically the lack of statistical significance testing, the limitation to CLIP/ViT architectures in the initial submission, and potential data leakage issues.

**Reviewer Concerns:**

The rebuttal successfully addressed the requests for mechanistic analysis and generalizability. Specifically, the authors provided t-SNE visualizations and classifier weight norm analyses to satisfy Reviewer yvXu's inquiries regarding the failure mechanisms of repeated under-sampling and the robustness of the proposed framework. They also conducted additional experiments on DINO to address concerns from Reviewers b51h and QeSQ regarding architectural limitations. Additionally, Reviewer QeSQ's concerns about data leakage and the instability of the LADE loss were resolved through experimental adjustments (moving IN21K results to the appendix) and technical clarifications.

 However, several issues remain outstanding. Reviewer WP1F maintained that the contribution lacks novelty compared to the LIFT baseline. Crucially, Reviewer b51h's request for broader architectural validation was not fully satisfied, as the authors acknowledged they did not conduct experiments on DINOv2, MAE, or SigLIP due to resource limitations or future planning. Furthermore, the authors did not address Reviewer b51h's concern regarding statistical significance testing (running multiple trials to report variance), opting instead to rely on fixed random seeds.

**Reviewer Scores:**

Reviewer yvXu (Score: 4) would likely maintain their score of 4 or increase the score. While the authors provided t-SNE visualizations and weight norm calculations , these are empirical observations rather than the "theoretical" or "geometric" analyses the reviewer explicitly requested. A strict reviewer would find the response descriptive rather than explanatory, as it failed to provide the mathematical or structural depth required to explain why the proposed framework is robust compared to the baselines.

Reviewer QeSQ (Score: 4) would also likely maintain their score of 4. Although the authors added DINO experiments , the explicit admission of omitting DINOv2 due to "GPU memory limitations"  is often considered an unacceptable justification for top-tier reproducibility standards. Furthermore, the explanation for the LADE loss collapse, dismissing it as "overly complex" and "suitable for CNNs rather than ViTs" without deeper diagnostic evidence, is subjective and lacks the rigorous ablation the reviewer sought regarding fundamental limitations versus hyperparameter sensitivity.

Reviewer b51h (Score: 6) would likely have maintained their score.While the addition of DINO experiments addressed generalization concerns, the authors failed to provide the requested statistical significance testing (variance), justifying the use of fixed seeds instead.

Reviewer WP1F (Score: 4) would definitively maintain their low score, as they explicitly stated in the post-rebuttal comments that despite the authors' arguments regarding training efficiency, the performance similarity to the LIFT baseline remains a critical flaw and the results are not yet satisfactory.

---

### Decision · Program_Chairs · 2026-01-26

Reject